# Impact of the Euro 2020 championship on the spread of COVID-19

Jonas Dehning [1,6], Sebastian B. Mohr [1,6], Sebastian Contreras [1], Philipp Dönges [1], Emil N. Iftekhar [1], Oliver Schulz [2], Philip Bechtle [3] ✉ & Viola Priesemann [1,4,5] ✉

Large-scale events like the UEFA Euro 2020 football (soccer) championship offer a unique opportunity to quantify the impact of gatherings on the spread of COVID-19, as the number and dates of matches played by participating countries resembles a randomized study. Using Bayesian modeling and the gender imbalance in COVID-19 data, we attribute 840,000 (95% CI: [0.39M, 1.26M]) COVID-19 cases across 12 countries to the championship. The impact depends non-linearly on the initial incidence, the reproduction number $R$, and the number of matches played. The strongest effects are seen in Scotland and England, where as much as 10,000 primary cases per million inhabitants occur from championship-related gatherings. The average match-induced increase in $R$ was 0.46 [0.18, 0.75] on match days, but important matches caused an increase as large as +3. Altogether, our results provide quantitative insights that help judge and mitigate the impact of large-scale events on pandemic spread.

Passion for competitive team sports is widespread worldwide. However, the tradition of watching and celebrating popular matches together may pose a danger to coronavirus disease 2019 (COVID-19) mitigation, especially in large gatherings and crowded indoor settings (see, e.g., refs. [1–6]). Interestingly, sports events taking place under substantial contact restrictions had only a minor effect on COVID-19 transmission[7–11]. However, large events with massive media coverage, stadium attendance, increased travel, and viewing parties can play a major role in the spread of COVID-19—especially if taking place in settings with few COVID-19-related restrictions. This was the case for the UEFA Euro 2020 Football Championship (Euro 2020 in short), staged from June 11 to July 11, 2021. While stadium attendance might only have a minor effect[12–14], it increases TV viewer engagement[15–17], and encourages additional social gatherings[18]. These phenomena and previous observational analyses[19] suggest that the Euro 2020's impact may have been considerable. Therefore, we used this championship as a case study to quantify the impact of large events on the spread of

COVID-19. Counting with quantitative insights on the impact of these events allows policymakers to determine the set of interventions required to mitigate it.

Two facts make the Euro 2020 especially suitable for the quantification. First, the Euro 2020 resembles a randomized study across countries: The time-points of the matches in a country do not depend on the state of the pandemic in that country and how far a team advances in the championship has a random component as well[20]. This independence between the time-points of the match and the COVID-19 incidence allows quantifying the effect of football-related social gatherings without classical biasing effects. This is advantageous compared to classical inference studies quantifying the impact of non-pharmaceutical interventions (NPIs) on COVID-19 where implementing NPIs is a typical reaction to growing case numbers[21–23]. Second, the attendance at match-related events, and thus the cases associated with each match, is expected to show a gender imbalance[24]. This was confirmed by news outlets and early studies[25–28]. Hence, the gender

[1]Max Planck Institute for Dynamics and Self-Organization, Am Faßberg 17, 37077 Göttingen, Germany. [2]Max Planck Institute for Physics, Föhringer Ring 6, 80805 München, Germany. [3]Physikalisches Institut, Universität Bonn, Nußallee 12, 53115 Bonn, Germany. [4]Institute for the Dynamics of Complex Systems, University of Göttingen, Friedrich-Hund-Platz 1, 37077 Göttingen, Germany. [5]Institute of Computer Science and Campus Institute Data Science, University of Göttingen, Goldschmidtstraße 7, 24118 Göttingen, Germany. [6]These authors contributed equally: Jonas Dehning, Sebastian B. Mohr. ✉e-mail: bechtle@physik.uni-bonn.de; viola.priesemann@ds.mpg.de

imbalance presents a unique opportunity to disentangle the impact of the matches from other effects on pathogen transmission rates.

Here we build a Bayesian model to quantify the effect large-scale sports events on the spread of COVID-19, using the Euro 2020 as case study. In the following, we use "case" to refer to a confirmed case of a severe acute respiratory syndrome coronavirus 2 (SARS-CoV-2) infection in a human and "case numbers" to refer to the number of such cases. Not all infections are detected and represented in the cases and cases come with a delay after the actual infection. Our model simulates COVID-19 spread in each country using a discrete renewal process[22,29] for each gender separately, such that the effect of matches can be assessed through the gender imbalance in case numbers. This is defined as "(male incidence – female incidence)/total incidence", and through the temporal association of cases to match dates of the countries' teams. Regarding the expected gender imbalance at football-related gatherings, we chose a prior value of 33% (95% percentiles [18%, 51%]) female participants, which is more balanced than the values reported for national leagues (about 20%)[24]. However, this agrees with the expected homogeneous and broad media attention of events like the Euro 2020. For the effective reproduction number $R_{eff}$ we distinguish three additive contributions; the base, NPI-, and behavior-dependent reproduction number $R_{base}$, a match-induced boost on it $\Delta R_{football}$, and a noise term $\Delta R_{noise}$, such that $R_{eff} = R_{base} + \Delta R_{football} + \Delta R_{noise}$. We assume $R_{base}$ to vary smoothly over time, while the effect of single matches $\Delta R_{football}$ is concentrated on one day and allows for a gender imbalance. The term $\Delta R_{noise}$ allows the model to vary the relative reproduction number for each gender independent of the football events smoothly over time. We analyzed data from all participating countries in the Euro 2020 that publish daily gender-resolved case numbers ($n = 12$): England, the Czech Republic, Italy, Scotland, Spain, Germany, France, Slovakia, Austria, Belgium, Portugal, and the Netherlands (ordered by resulting effect size). We retrieved datasets directly from governmental institutions or the

COVerAGE-DB[30]. See Supplementary Section S1 for a list of data sources. Our analyses were carried out following FAIR[31] principles; all code, including generated datasets, are publicly available (https://github.com/Priesemann-Group/covid19_soccer).

## Results

### The main impact arises from the subsequent infection chains

We quantified the impact of the Euro 2020 matches on the reproduction number for the 12 analyzed countries (Fig. 1a) and for every single match (Supplementary Fig. S8). On average, a match increases the reproduction number $R$ by 0.46 (95% CI [0.18, 0.75]) (Fig. 1a and Supplementary Table S4) for a single day. In other words, when a country participated in a match of the Euro 2020 championship, every individual of the country infected on average $\Delta R_{match}$ extra persons (see Supplementary Section S2 for more details). The cases resulting from these infections occurring at gatherings on the match days are referred to as primary cases.

However, primary cases are only the tip of the iceberg; any of these cases can initiate a new infection chain, potentially spreading for weeks (see Supplementary Section S2 for more details). We included all subsequent cases until July 31, which is about two weeks after the final. As expected, subsequent cases outnumber the primary cases considerably at a ratio of about 4:1 on average (Supplementary Table S3). As a consequence, on average, only 3.2% [1.3%, 5.2%] of new cases are directly associated with the match-related social gatherings throughout that analysis period (Fig. 1b). This surge of subsequent cases highlights the long-lasting impact of potential single events on the COVID-19 spread (see Supplementary Table S2).

We find an increase in COVID-19 spread at the Euro 2020 matches in all countries we analyzed, except for the Netherlands. In the Netherlands, a "freedom day" coincided with the analysis period[32] and was accompanied by the opposite gender imbalance compared to the football matches, thereby apparently inverted the football effect.

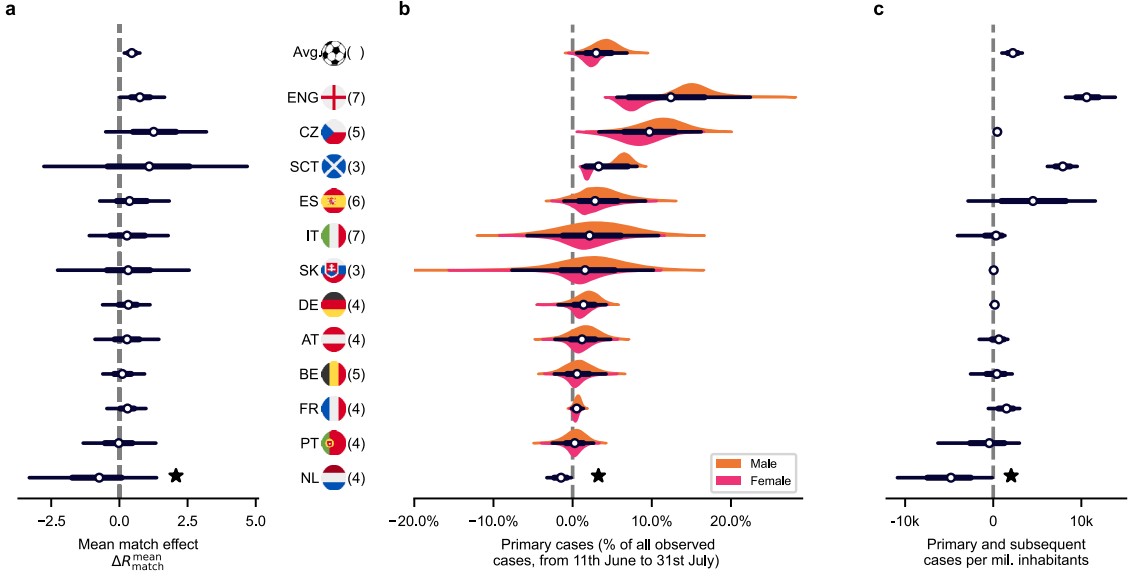

**Fig. 1 | Quantifying the impact of the Euro 2020 on COVID-19 spread. a** Using Bayesian inference and an SEIR-like model, we infer the mean increase on the reproduction number associated with Euro 2020 matches, $\Delta R_{match}^{mean}$, in each analyzed country ($n = 12$ countries). Almost all countries show a median of the mean increase larger than zero (cf. Supplementary Table S4). Note that in the Netherlands (★) a complete lifting of restrictions was implemented on June 26 2021 ("freedom day"). Apparently, its impact also had the opposite gender imbalance, making it hard for the model to extract the Euro 2020's effect (Supplementary Fig. S31). **b** The $\Delta R_{match}^{mean}$ enables us to quantify the primary cases, i.e., cases associated directly with the match days (as percentage of all cases from June 11 to July 31 2021). **c** Any

primary infection at a match can start an infection chain. The total number of primary and subsequent cases that were inferred to be causally related to the Euro 2020 from its start until 31 July depend on the COVID-19 prevalence and the base spread during the analysis period. In parentheses are the number of matches played by the respective team. White dots represent median values, black bars and whiskers correspond to the 68% and 95% credible intervals (CI), respectively, and the distributions in color (truncated at 99% CI) represent the differences by gender (Supplementary Table S2). The Netherlands is left out from the average calculations and subsequent analyses.

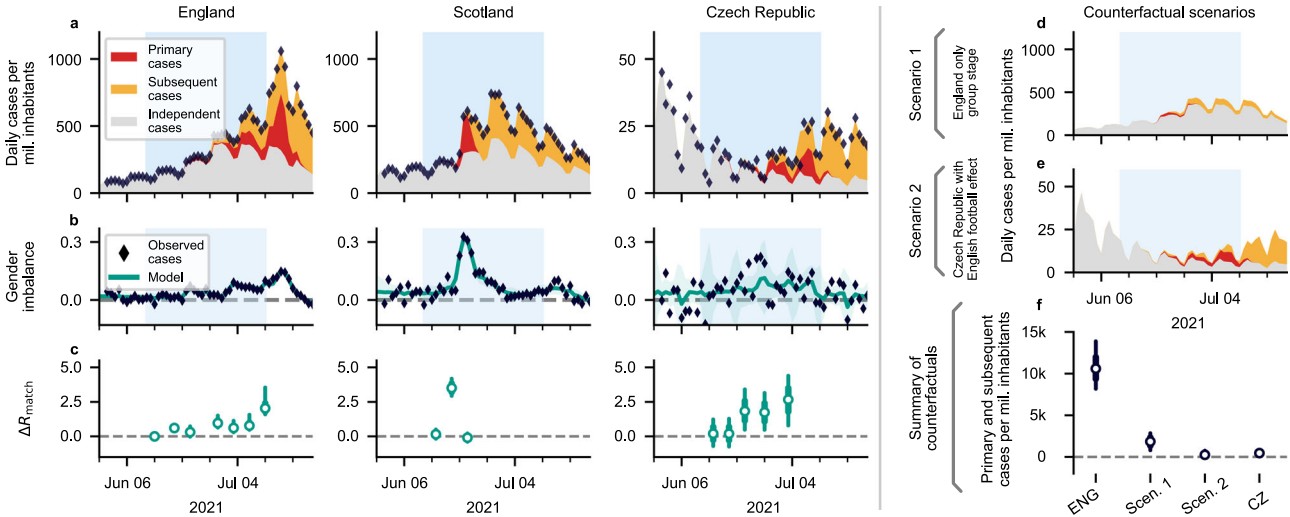

**Fig. 2 | Example cases illustrate that the spread associated with the Euro 2020 can encompass a substantial fraction of the observed cases. a** The model enables one to split the observed incidence (black diamonds) into: cases independent of Euro 2020 matches (gray area), primary cases (directly associated with Euro 2020 matches, red area), and subsequent cases (additional infection chains started by primary cases, orange area). See Supplementary Information for all countries (Supplementary Figs. S24–S36). Here and in all following figures, the light blue shaded area signifies the time span of the Euro 2020. **b** Football-related gatherings, and hence the case numbers, show a gender imbalance. This facilitates the inference of the football-related increase in COVID-19 spread. Here the turquoise shaded areas correspond to 95% CI. **c** The effect of social gatherings at match days is modeled as a single additive increase in the reproduction number $\Delta R_{match}$ concentrated on the day of each match. For example, $\Delta R_{match} = 2$ means that, on the day of the match, each infected individual on average infected two additional persons (on top of the base trend). **d, e** The counterfactual scenario assumes that England would not have reached the knockout phase (**d**, Scen. 1), or that the Czech fans and matches would have been equal to the English (i.e., reaching the final, and Czech people doing the same football-related gatherings as the English by their impact on disease spread; **e**, Scen. 2). **f** In the counterfactual scenarios, the Euro 2020 would have had much smaller impact with fewer matches (Scen. 1), or with an overall more favorable pandemic situation as in the Czech Republic (Scen. 2). White dots represent median values, bars and whiskers correspond to the 68% and 95% credible intervals (CI).

Therefore, we exclude the Netherlands from general averages and correlation studies, but still display the results for completeness.

The primary and subsequent cases on average amounted to 2200 (95% CI [986, 3308]) cases per million inhabitants (Fig. 1c and Supplementary Table S2). This amounts to about 0.84 million (CI: [0.39M, 1.26M]) cases related to the Euro 2020 in the 12 countries (cf. Supplementary Table S3). With the case fatality risk of that period, this corresponds to about 1700 (CI: [762, 2470]) deaths, assuming that the primary and subsequent spread affects all ages equally. Most likely this is slightly overestimated since the age groups most at risk from COVID-19-related death are probably underrepresented in football-related social activities and thus more unlikely to be affected by primary championship-related infections. However, the overall number of primary and subsequent cases attributed to the championship is dominated by the subsequent cases, and the mixing of individuals of different age-groups then mitigates this bias. Individually, three countries, England, the Czech Republic, and Scotland showed a significant increase in COVID-19 incidence associated with the Euro 2020, and Spain and France show an increase at the one-sided 90% significance threshold. In other countries such as Germany, only a relatively small contribution of primary cases was associated with the Euro 2020 championship, and a small gender imbalance was observed. Low COVID-19 incidence during the championship or imprecise temporal association between infection and confirmation of it as a case can lead to a loss of sensitivity and hinder the detection of an effect, as can be seen from the large width of several posterior distributions (e.g., Italy and Slovakia, which had particularly low incidence).

### The strongest effect is observed in England and Scotland

Overall, the effect of the Euro 2020 was quite diverse across the participating countries, ranging from almost no additional infections to up to 1% of the entire population being infected (i.e., from Portugal to England, Fig. 1). To illustrate this diversity, the comparison between England, Scotland, and the Czech Republic is particularly illustrative

(Fig. 2). For all countries, we disentangled the cases that are considered to happen independently of the Euro 2020 (Fig. 2a, gray), the primary cases directly associated with gatherings on the days of the matches (red), and the subsequent infection chains started by the primary cases (orange; see Supplementary Figs. S24–S36 for all countries).

England, being the runner-up of the championship and thus played the maximum number of matches, displays the strongest effect over the longest duration, with a substantial increase in reproduction number $\Delta R_{match}$ towards the last matches of the championship. This reflects the increasing popularity of the later matches, as e.g., quantified by the increase of the search term on Google (Supplementary Fig. S20). Scotland shows a particularly strong effect of a single match (Scotland vs England) staged in London during the group phase, with $\Delta R_{match} = 3.5$ [2.9, 4.2] (Fig. 2c). This means that on average over the total Scottish population, every single person infected additional 3.5 persons at or around that single day. These are very strong effects. As a consequence, in Scotland the subsequent cases from the single match accounted for about 30% of the cases in the following weeks, illustrating the impact of such gatherings on public health.

### Low overall incidence prevents large match-related spread

In the Czech Republic, the situation was different compared to England and Scotland, although the analyses point to similarly strong gatherings on the match days (i.e., large $\Delta R_{match}$, Fig. 1a). However, because of the overall low incidence much fewer people were infected throughout the championship. The advantage of low incidence or fewer games is illustrated in two counterfactual scenarios. Even under the assumption that the Czech team had continued to the final and the population had gathered exactly like the English (i.e., showing the same $\Delta R_{match}$ in the matches they played), the total number of cases (per million) would have been more than 40 times lower than in England, owing to the lower base incidence and a lower base reproduction number (Fig. 2d). Assuming, as a counterfactual scenario, that England had dropped out in the group stage, the number of cases associated with the Euro 2020

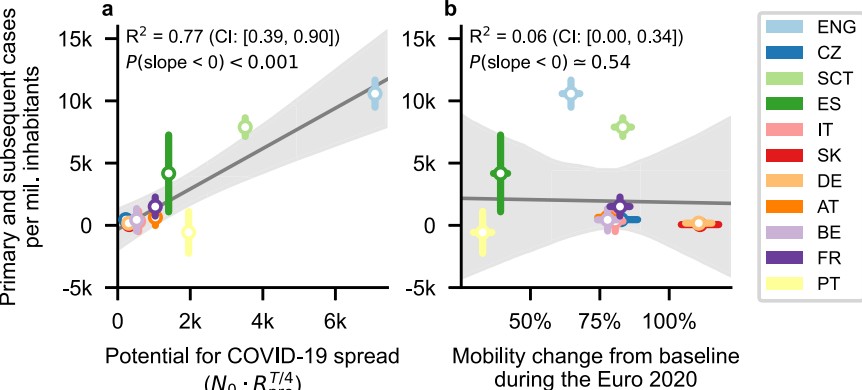

**Fig. 3 | Which variables can predict the extent of the impact of Euro 2020 matches? a** The potential for spread, i.e., the number of COVID-19 cases that would be expected during the time $T$ a country is playing in the Euro 2020 ($N_0 \cdot R_{pre}^{T/4}$), is strongly correlated with the number of Euro 2020-related cases. Therefore, policymakers should simultaneously consider the initial incidence $N_0$, reproduction number prior to the event $R_{pre}$, and expected duration of an event $T$ to assess whether it is pertinent to allow it (The correlation is not significant if England and Scotland are left out, but the slope is still consistent with this result.). **b** Mobility changes from baseline during the Euro 2020 are not correlated with the number of COVID-19 cases associated with the championship in each country. Furthermore, the direction of the effect of mobility per se in this context is unclear. The gray line and area are the median and 95% CI of the linear regression ($n = 11$ countries; The Netherlands was excluded for this analysis). Whiskers denote one standard deviation.

would have been much lower. This suggests that both the success in the championship and the base incidence and behavior in a country influence the public health impact of such large-scale events.

To better understand the impact of the Euro 2020, we quantified the determinants of the spread across countries. From theory, we expect the absolute number of infections generated by Euro 2020 matches to depend non-linearly on a country's base incidence $N_0$, which determines the probability to meet an infected person, and on the effective reproduction number prior to the championship $R_{pre}$, as a gauge for the underlying infection dynamics generating the subsequent cases, which determines how strongly an additional infection spreads in the population. We can then define the potential for COVID-19 spread as the number of COVID-19 cases that would be expected during the time $T$ a country is playing in the Euro 2020 ($N_0 \cdot R_{pre}^{T/4}$), assuming a generation interval of 4 days. Indeed, we find a clear correlation between the observed and the expected incidence Fig. 3a, $R^2 = 0.77$ (95% CI [0.39, 0.9]), $p < 0.001$, with a slope of 1.62 (95% CI [1.0, 2.26]). The strong significance of this correlation relies mainly on England and Scotland. However, the observed slope in an analysis without these two countries (0.76, 95% CI: [−1.46, 3.04]), while not significant at the 95% confidence level, is consistent with the findings including all countries. This is shown in Supplementary Fig. S7.

Furthermore, quantifying correlations between $N_0$ and $R_{pre}$ and the number of primary and subsequent cases related to the Euro 2020, we see a trend for each (Supplementary Fig. S6a, b). However, these are weak and statistically significant only for $R_{pre}$. Altogether, our data suggest that a favorable pandemic situation (low $R_{pre}$ and low $N_0$) before the gatherings, and low $R_{base}$ during the period of gatherings jointly minimize the impact of the Euro 2020 on community contagion. A prerequisite for this is that the known preventive measures, such as reducing group size, imposing preventive measures, and minimizing the number of encounters remain encouraged.

Independently on the epidemic situation, Euro 2020's effect might be influenced by people's prudence and the team's popularity and success during the championship. While we do not observe any obvious effect of local mobility as a measure of the prudence of people (Fig. 3b, $R^2 = 0.06$ (95% CI [0.00, 0.34]), $p = 0.54$, and Supplementary Fig. S4), the potential popularity −represented by the number of matches played and hosted by a given country− had a more notable trend (Supplementary Fig. S6c). Still, this correlation was not statistically significant. Moreover, we found no relationship between the effect size and the Oxford governmental response tracker[33] (Supplementary Fig. S5).

## Discussion

Large international-scale sports events like the Euro 2020 Football Championship have the potential to gather people like no other type of event. Our quantitative insights on the impact of such gatherings on COVID-19 spread provide policymakers with tools to design the portfolio of interventions required for mitigation (using, e.g., results of refs. 22,23,34). Thereby, our quantification can support society in carefully weighing the positive social, psychological, and economic effects of mass events against the potential negative impact on public health[35]. Our analysis attributes about 0.84 million (95% CI: [0.39M, 1.26M]) additional infected persons to the Euro 2020 championship. Assuming that the primary and subsequent spread affects all ages equally, this corresponds across the 12 countries to about 1700 (CI: [762, 2470]) deaths. Thus, the public health impact of the EURO 2020 was not negligible.

To prevent the impacts of these events, measures, such as promoting vaccination, enacting mask mandates, and limiting gathering sizes, can be helpful. Besides, the effectiveness of such interventions has already been quantified in different settings (e.g., refs. 22,23) so that policymakers can weigh them according to specific targets and priorities. Furthermore, focused measures that aim to mitigate disease spread in situ, such as testing campaigns and requiring COVID passports to attend sport-related gatherings and viewing parties, present themselves as helpful options. In addition, one could encourage participants of a large gathering to self-quarantine and test themselves afterward. Moreover, the championship distribution of matches every 4−5 days coincides with the mean incubation period and generation interval of COVID-19. This means that individuals who get infected watching a match can turn infectious by the subsequent while potentially pre-symptomatic. Such resonance effects between gathering intervals and incubation time can increase the spread considerably[34]. It thus depends on the design of the championships, on the precautionary behavior of individuals, and on the basic infection situation how much large-scale events threaten public health, even if the reproduction number is transiently increased during these events.

Previous studies that evaluated the impact of sports events on the spread of COVID-19 and considered the spectator gatherings at match venues were not conclusive[7,8,36]. This agrees with our results as we find the impact of hosting a match to be small to non-existent (Supplementary Fig. S9). However, location having little effect may well be specific to the Euro 2020, where matches were distributed across different countries. In the traditional settings of the UEFA European

Football Championship or the FIFA World Cup, a single country or a small group of countries hosts the entire championship, and the championship is accompanied by elaborate supporting events, public viewing, and extensive travel of international guests. Hence, for other championships, such as, e.g., the FIFA World Cup 2022 in Qatar or the Euro 2024 in Germany, the impact of location might be considerably larger.

Our model accounts for slow changes in the transmission rates that are unrelated to football matches through the gender-independent reproduction number $R_{base}$. We find $R_{base}$ to increase at least transiently during the championship in all 12 countries except for England and Portugal (Supplementary Figs. S24–S35). The above may suggest that our estimate of the match effect $\Delta R_{match}$ is conservative: The overall increase of COVID-19 spread might in part be attributed to $R_{base}$, but will not be incorrectly associated with football matches. Our results might further be biased if the incidence and the teams' progression in the Euro 2020 are correlated. It is conceivable that high incidence would negatively correlate with team progression through ill or quarantined team members. However, there were only few such cases during the Euro 2020[37], and the correlation might also be positive: At higher case numbers the team might be more careful. Hence, the correlation is unclear and probably negligible.

The COVID-19 spread obviously depends on many factors. However, many of those parameters, such as the vaccination rate, the contact behavior or motivation to be tested, are changing slowly over time and hence can be absorbed into the slowly changing base reproduction rate $R_{base}$ and the gender-asymmetric noise $\Delta R_{noise}$; other parameters, like social and regional differences, age-structure or specific contact networks are expected to be constant over time and average out across a country. To further test the robustness of our model, we systematically varied the prior assumptions on the central model parameters, among them the delay (Supplementary Fig. S12), the width of the delay kernel (Supplementary Fig. S13), the change point interval (Supplementary Fig. S14), the generation interval (Supplementary Fig. S16) and a range of other priors (Supplementary Fig. S17). Furthermore, when using wider prior ranges for the gender imbalance, football-related COVID-19 cases remain unchanged but the uncertainty increases (Supplementary Fig. S15), thus validating our choice. Even for the case of prior symmetric gender imbalance assumptions, the posterior distribution of the female participation converges for the three most significant countries to median values between 20 and 45%. As last cross-check, we made sure that we found no effect when shifting the match dates by 2 weeks relative to the case numbers (Supplementary Fig. S10) nor by shifting match dates outside the championship range, by more than ±30 days (Supplementary Fig. S11).

Besides quantifying the impact of matches on the reproduction number, our methodology allowed us to estimate the delay between infections and confirmation of positive tests $D$ without a requirement to identify the source of each infection (Supplementary Fig. S19). Our estimates for $D$ in the participating countries were around 3-5 days (England: 4.5 days (95% CI [4.3, 5]), Scotland: 3.5 days (95% CI [3.3, 3.8]), Supplementary Figs. S24 and S33g and Supplementary Table S4). This agrees with available literature and is an encouraging signal for the feasibility of containing COVID-19 with test-trace-and-isolate[38–42]. However, we expect that some individuals would actively get tested right after a match, thereby increasing the case finding and reporting rates. This can slightly affect our estimates for the delay distribution $D$ and would require additional information to be corrected. Altogether, analyzing large-scale events with precise timing and substantial impact on the spread presents a promising, resource-efficient complement to classical quantification of delays.

Understanding how popular events with major in-person gatherings affect the spreading dynamics of COVID-19 can help us design better strategies to prevent new outbreaks. The Euro 2020 had a pronounced impact on the spread despite considerable awareness of the risks of COVID-19. We estimate that, e.g., about 48% of all cases in England until July 31 are related to the championship. In future, with declining awareness about COVID-19 but potentially better immunity, similar mass events, such as the football world cups, the Super Bowl, or the Olympics, will still unfold their impact. Acute, long-COVID-19 and post-COVID-19 will continue to pose a challenge to societies in the years to come. Our analysis suggest that a combination of low $R_{pre}$ and low initial incidence at the beginning of the event, together with the known preventive measures, can strongly reduce the impact of these events on community contagion. Fulfilling these preconditions and increasing health education in the general population can substantially reduce the adverse health effects of future mass events.

## Methods

To estimate the effect of the championship in different countries, we constructed a Bayesian model that uses the reported case numbers in 12 countries. Ethical approval was not sought as we only worked with openly available data. A graphical overview of the inference model is given in Fig. 4 and model variables, prior distributions, indices, country-dependent priors, and sampling performance are summarized in Tables 1, 2, 3, 4 and 5, respectively.

### Modeling the spreading dynamics, including gender imbalance

The model simulates the spread of COVID-19 in each country separately using a discrete renewal process[22,29,43]. We infer a time-dependent effective reproduction number with gender interactions between genders $g$ and $g'$, $R_{eff,g,g'}(t)$, for each country[21].

Even though participation of women in football fan activity has increased in the last decades[44], football fans are still predominantly male[24]. Hence one expects a higher infection probability at the days of the match for the male compared to the female population. Integrating this information into the model by using gender resolved case numbers, allows improved inference of the Euro 2020's impact. In the following, genders "male" and "female" are denoted by the subscripts $\bullet_{g=1}$ and $\bullet_{g=2}$, respectively. Furthermore, we modeled the spreading dynamics of COVID-19 in each country separately.

In the discrete renewal process for disease dynamics of the respective country, we define for each gender $g$ a susceptible pool $S_g$ and an infected pool $I_g$. With $N$ denoting the population size, the spreading dynamics with daily time resolution $t$ reads as

$$I_g(t) = \frac{S_g(t)}{N} \sum_{g'=1}^{2} \mathbf{R}_{eff,g,g'}(t) \sum_{\tau=0}^{10} I_{g'}(t - 1 - \tau)\, G(\tau), \quad (1)$$

$$S_g(t) = S_g(t-1) - E_g(t-1), \quad (2)$$

$$G(\tau) = \text{Gamma}(\tau; \mu = 4, \sigma = 1.5). \quad (3)$$

We apply a discrete convolution in Eq. (1) to account for the latent period and subsequent infection (red box in Fig. 4). This generation interval (between infections) is modeled by a Gamma distribution $G(\tau)$ with a mean $\mu$ of four days and standard deviation $\sigma$ of one and a half days. This is a little longer than the estimates of the generation interval of the Delta variant[45,46], but shorter than the estimated generation interval of the original strain[47,48]. The impact of the choice of generation interval has negligible impact on our results (Supplementary Fig. S16). The infected compartment (commonly $I$) is not modeled explicitly as a separate compartment, but implicitly with the assumed generation interval kernel.

The effective spread in a given country is described by the country-dependent effective reproduction numbers for infections of

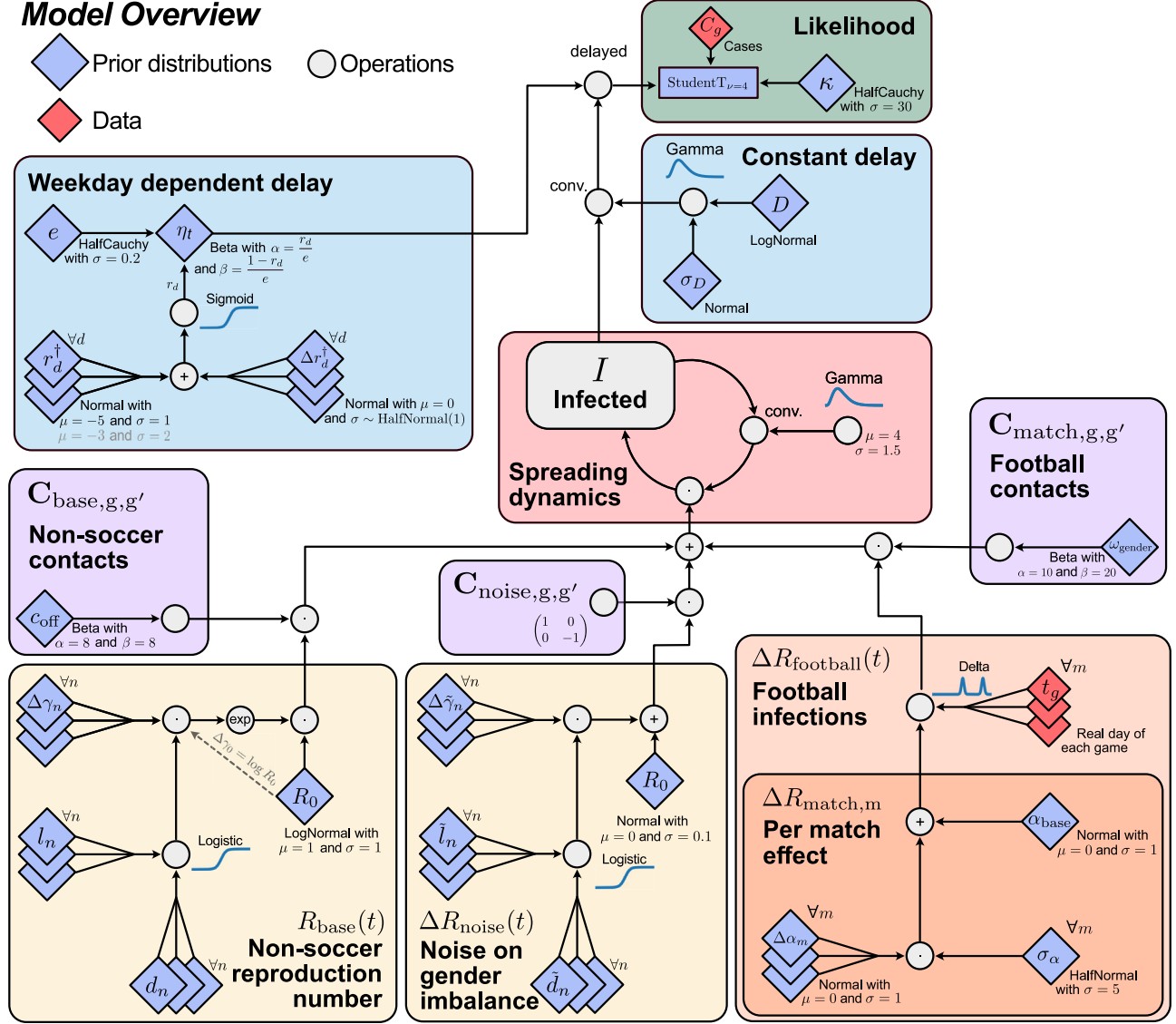

**Fig. 4 | Model overview illustrating the relationship between the chosen prior distributions and the disease dynamics.** Boxes in the flowchart are color-coded according to what they describe. Light blue boxes: delay modulations. Green boxes: likelihoods. Red boxes: spreading dynamics. Purple boxes: contact matrices. Yellow boxes: effects independent of football matches. Orange boxes: effects of the football matches. Diamonds show prior distributions (blue) or incorporated data (red), and gray circles denote any mathematical operation.

individuals of gender $g$ by individuals of gender $g'$

$$\mathbf{R}_{\mathrm{eff},g,g'}(t) = R_{\mathrm{base}}(t)C_{\mathrm{base},g,g'} + \Delta R_{\mathrm{football}}(t)C_{\mathrm{match},g,g'} + \Delta R_{\mathrm{noise}}(t)C_{\mathrm{noise},g,g'},$$
(4)

where $C_{\mathrm{base},g,g'}$, $C_{\mathrm{match},g,g'}$, and $C_{\mathrm{noise},g,g'}$ describe the entries of the contact matrices $\mathbf{C}_{\mathrm{base}}$, $\mathbf{C}_{\mathrm{match}}$, $\mathbf{C}_{\mathrm{noise}}$ respectively (purple boxes in Fig. 4).

This effective reproduction number is a function of three different reproduction numbers (yellow and orange boxes in Fig. 4):

1. A slowly changing base reproduction number $R_{\mathrm{base}}$ (22) that has the same effect on both genders; besides incorporating the epidemiological information given by the basic reproduction number $R_0$, it represents the day-to-day contact behavior, including the impact of non-pharmaceutical interventions (NPIs), voluntary preventive measures, immunity status, etc.

2. The reproduction number associated with social gatherings in the context of a football match $R_{\mathrm{match}}(t)$ (11); this number is only different from zero on days with matches that the respective

country's team participates in and it has a larger effect on men than on women.

3. A slowly changing noise term $\Delta R_{\mathrm{noise}}(t)$ (31), which subsumes all additional effects which might change the incidence ratio between males and females (gender imbalance).

The interaction between persons of specific genders is implemented by effective contact matrices $\mathbf{C}_{\mathrm{match}}$, $\mathbf{C}_{\mathrm{base}}$ and $\mathbf{C}_{\mathrm{noise}}$. All three are assumed to be symmetric.

$\mathbf{C}_{\mathrm{base}}$ describes non-football related contacts outside the context of Euro 2020 matches (left purple box in Fig. 4):

$$\mathbf{C}_{\mathrm{base}} = \begin{pmatrix} 1 - c_{\mathrm{off}} & c_{\mathrm{off}} \\ c_{\mathrm{off}} & 1 - c_{\mathrm{off}} \end{pmatrix},$$
(5)

with $c_{\mathrm{off}} \sim \mathrm{Beta}(\alpha = 8, \beta = 8)$.
(6)

**Table 1 | The intermediate variables of the model and their meaning**

| Variable | Meaning | Equation |
|---|---|---|
| $R_{\mathrm{eff},g,g'}(t)$ | Effective reproduction number between genders $g$ and $g'$ | (4) |
| $S_g(t)$ | Number of susceptible persons of gender $g$ | (2) |
| $I_g(t)$ | Number of infected persons of gender $g$ | (1) |
| $N$ | Population size | |
| $G(\tau)$ | Generation interval (Gamma kernel) | (3) |
| $R_{\mathrm{base}}(t)$ | Base reproduction number | (22) |
| $\Delta R_{\mathrm{football}}(t)$ | Time dependent additive reproduction number due to football matches | (11) |
| $\Delta R_{\mathrm{noise}}(t)$ | Time dependent additive reproduction number due other non-balanced transmission | (31) |
| $\Delta R_{\mathrm{match},m}$ | Additive reproduction number of match $m$ | (13) |
| $\mathbf{C}_{\mathrm{base}}$ | Base contact matrix between genders | (5) |
| $\mathbf{C}_{\mathrm{match}}$ | Contact matrix for football related gatherings | (8) |
| $\mathbf{C}_{\mathrm{noise}}$ | Contact matrix for other non-balanced transmission | (10) |
| $t_m$ | Day of match $m$ | |
| $\alpha_{\mathrm{prior}}$ | Vector encoding country participation in matches | |
| $\beta_{\mathrm{prior}}$ | Vector encoding whether country hosted matches | |
| $\Delta\alpha_m$ | Difference of the effect of individual matches $m$ (the country participated in) to mean effect of such matches | (15) |
| $\Delta\beta_m$ | Difference of the effect of individual matches $m$ (the country hosted) to mean effect of such matches | (20) |
| $\gamma_n(t)$ | Time-dependent base reproduction number in log-space between change point $n$ and $n+1$ | (24) |
| $\Delta\gamma_n$ | Effect of the change point $n$ | (25) |
| $\tilde{\gamma}_n(t)$ | Additive reproduction number due other non-balanced transmission between change point $n$ and $n+1$ | (33) |
| $\Delta\tilde{\gamma}_n$ | Effect of the change point $n$ on the non-balanced transmission | (34) |
| $C_g^{\dagger}(t)$ | Delayed number of infected persons of gender $g$ | (40) |
| $\bar{D}_{\mathrm{country}}$ | Country dependent reporting delay | Table 4 |
| $\hat{C}_g(t)$ | Modeled number of cases of gender $g$ | (44) |
| $\eta_t$ | Fraction of daily delayed cases | (45) |
| $r_d$ | Average value of the fraction of delayed cases on weekday $d$ | (46) |
| $r_d^{\dagger}$ | Logit-transformed average value of the fraction of delayed cases on weekday $d$ | (47) |
| $\Delta r_d^{\dagger}$ | Deviation from the prior value of the fraction of delayed cases on weekday $d$ | (50) |
| $C_g(t)$ | Measured number of cases of gender $g$ | (53) |
| $R_{\mathrm{pre}}$ | Reproduction number two weeks prior to the start of the Euro 2020 | Used in Fig. 3 |
| $I_{\mathrm{primary}}$ | Number of primary infected persons due to football matches | (65) |
| $I_{\mathrm{subsequent}}$ | Number of subsequent infected persons due to football matches | (67) |
| $I_{\mathrm{none}}$ | Number of infected persons without considering football matches | (67) |

Here, we have the prior assumption that contacts between women, contacts between men, and contacts between women and men are equally probable. Hence, we chose the parameters for the Beta distribution such that $c_{\mathrm{off}}$ has a mean of 50% with a 2.5th and 97.5th percentile of [27%, 77%]. This prior is chosen such that it is rather uninformative. As shown in Supplementary Fig. S17, this and other priors of auxiliary parameters do not affect the parameter of interest if their width is varied within a factor of 2 up and down.

$\mathbf{C}_{\mathrm{match}}$ describes the contact behavior in the context of the Euro 2020 football matches (right purple box in Fig. 4). Here, we assume as a prior that the female participation in football-related gatherings accounts for $\simeq 33\%$ (95% percentiles [18%,51%]) of the total participation. Hence, we get the following contact matrix

$$\mathbf{C}_{\mathrm{match,unnorm.}} = \begin{pmatrix} (1-\omega_{\mathrm{gender}})^2 & \omega_{\mathrm{gender}}(1-\omega_{\mathrm{gender}}) \\ \omega_{\mathrm{gender}}(1-\omega_{\mathrm{gender}}) & \omega_{\mathrm{gender}}^2 \end{pmatrix} \quad (7)$$

$$\mathbf{C}_{\mathrm{match}} = \frac{\mathbf{C}_{\mathrm{match,unnorm.}}}{|\mathbf{C}_{\mathrm{match,unnorm.}} \cdot (0.5,0.5)^T|_2} \quad (8)$$

$$\omega_{\mathrm{gender}} \sim \mathrm{Beta}(\alpha=10, \beta=20). \quad (9)$$

The prior beta distribution of $\omega_{\mathrm{gender}}$ is bounded between at 0 and 1 and with the parameter values of $\alpha=10$ and $\beta=20$ has the expectation value of 1/3. The robustness of the choice of this parameter is explored in Supplementary Fig. S15. $\mathbf{C}_{\mathrm{match}}$ is normalized such that for balanced case numbers (equal case numbers for men and women) and an additive reproduction number $R_{\mathrm{match}}=1$ will lead to a unitary increase of total case numbers. The reproduction number of women will therefore increase by $2\omega_{\mathrm{gender}}\Delta R_{\mathrm{match}}(t)$ on match days whereas the one of men will increase by $2(1-\omega_{\mathrm{gender}})\Delta R_{\mathrm{match}}$, assuming balanced case numbers beforehand.

$\mathbf{C}_{\mathrm{noise}}$ describes the effect of an additional noise term, which changes gender balance without being related to football matches (middle purple box in Fig. 4). For simplicity, it is implemented as

$$\mathbf{C}_{\mathrm{noise}} = \begin{pmatrix} 1 & 0 \\ 0 & -1 \end{pmatrix}, \quad (10)$$

whereby we center the diagonal elements such that the cases introduced by the noise term sum up to zero, i.e. $\sum_{i,j} R_{\mathrm{noise}} \cdot C_{\mathrm{noise},i,j} = 0$.

### Football-related effect

Our aim is to quantify the number of cases (or equivalently the fraction of cases) associated with the Euro 2020, $\Gamma_g^{\mathrm{Euro}}$. To that end we assume

## Table 2 | Prior distributions

| Variable | Meaning | Prior distribution | Equation |
|---|---|---|---|
| $c_{off}$ | Off-diagonal term of non-football related interaction matrix | $\text{Beta}(\alpha=8, \beta=8)$ | (6) |
| $\omega_{gender}$ | The fraction of female participation in football related gatherings compared to the total participation | $\text{Beta}(\alpha=10, \beta=20)$ | (9) |
| $\Delta R_{match}^{mean}$ | Mean gathering-related match effect | $\mathcal{N}(\mu=0, \sigma=5)$ | (14) |
| $\Delta R_{stadium}^{mean}$ | Mean effect of hosting a match at the stadium | $\mathcal{N}(\mu=0, \sigma=5)$ | (19) |
| $\sigma_\alpha$ | Prior value of the deviation from the mean match effect | $\text{HalfNormal}(\sigma=5)$ | (16) |
| $\sigma_\beta$ | Prior value of the deviation from the mean stadium effect | $\text{HalfNormal}(\sigma=5)$ | (21) |
| $R_O$ | Value of $R_{base}(t)$ at $t=0$ | $\text{LogNormal}(\mu=1, \sigma=1)$ | (23) |
| $\sigma_{\Delta\gamma}$ | Prior value of the effect of the change points of the base reproduction number | $\text{HalfCauchy}(0.5)$ | (26) |
| $l_n$ | Length of the change point $n$ | $\log(1 + \exp(\mathcal{N}(4,1)))$ | (27) |
| $d_n$ | Date of the change point $n$ | 27th May 2021 $+ 10 \cdot n + \mathcal{N}(0,3.5)$ | (29) |
| $\Delta R_{O,noise}$ | Value of $\Delta R_{noise}(t)$ at $t=0$ | $\mathcal{N}(\mu=0, \sigma=0.1)$ | (32) |
| $\sigma_{\Delta\tilde{\gamma}}$ | Prior value of the effect of the change points of the reproduction number of other non-balanced transmission | $\text{HalfCauchy}(0.2)$ | (35) |
| $\tilde{l}_n$ | Length of the non-balanced transmission change point $n$ | $\log(1 + \exp(\mathcal{N}(4,1)))$ | (36) |
| $\tilde{d}_n$ | Date of the non-balanced transmission change point $n$ | 27th May 2021 $+ 10 \cdot n + \mathcal{N}(0,3.5)$ | (38) |
| $D$ | Median of the latent period and reporting delay kernel | $\log(\mathcal{N}(\mu=\exp(\bar{D}_{country}), \sigma=\sigma_{\log\bar{D}}))$ | (41) |
| $\sigma_D$ | Standard deviation of the delay kernel | $\mathcal{N}(\mu=0.2\cdot\bar{D}_{country}, \sigma=0.08\cdot\bar{D}_{country})$ | (43) |
| $r_{base,d}^\dagger$ | Prior fraction of the logit-transformed weekday dependent delay | | (48), (49) |
| $\sigma_r$ | Prior deviation of the different weekdays from the prior of the fraction of delayed cases | $\text{HalfCauchy}(1)$ | (51) |
| $e$ | Prior deviation of each day from the weekday dependent delay | $\text{HalfCauchy}(0.2)$ | (52) |
| $\kappa$ | Overdispersion of the observed cases around the expected number of cases | $\text{HalfCauchy}(20)$ | (54) |

These are all the prior distributions and their meaning in our main model.

## Table 3 | Indices

| Index | Meaning | Values |
|---|---|---|
| $\cdot_g$ | Gender | 1 = male; 2 = female |
| $\cdot_m$ | Match | |
| $\cdot_n$ | Change point | |
| $\cdot_t$ | Time (in days) | |
| $\cdot_d$ | Weekday | Monday, ..., Sunday |

We use these standardized indices in our model.

## Table 4 | Country-dependent priors on the delay structure

| Country | Reporting convention | Prior delay ($\bar{D}_{country}$) | Scale of prior delay ($\sigma_{\log\bar{D}}$) |
|---|---|---|---|
| England | Symptom onset | 4 days | 0.1 |
| Scotland | Symptom onset | 4 days | 0.1 |
| Germany | Reporting date | 7 days | 0.1 |
| France | Symptom onset | 4 days | 0.1 |
| Austria | Unknown | 5 days | 0.15 |
| Belgium | Unknown | 5 days | 0.15 |
| The Czech Republic | Unknown | 5 days | 0.15 |
| Italy | Unknown | 5 days | 0.15 |
| The Netherlands | Symptom onset | 4 days | 0.1 |
| Portugal | Unknown | 5 days | 0.15 |
| Slovakia | Unknown | 5 days | 0.15 |
| Spain | Unknown | 5 days | 0.15 |

These priors depend on the definition of the date in the daily case numbers, which for some countries refers to symptom onset, sample collection or sample analysis.

that infections can occur at public or private football screenings in the two countries participating in the respective match $m$ (parameterized by $\Delta R_{match,m}$). Note that for the Euro 2020 not a single country, but a set of 11 countries hosted the matches. The participation of a team or the staging of a match in a country may have different effect sizes. Thus, we define the football related additive reproduction number as

$$\Delta R_{football}(t) = \sum_m \Delta R_{match,m} \cdot \delta(t_m - t). \qquad (11)$$

We assume the effect of each match to only be effective in a small time window centered around the day of a match $m$, $t_m$ (light orange box in Fig. 4). Thus, we apply an approximate delta function $\delta(t_m - t)$. To guarantee differentiability and hence better convergence of the model, we did not use a delta distribution but instead a narrow normal distribution centered around $t_m$, with a standard deviation of one day:

$$\delta(t) = \frac{1}{\sqrt{2\pi}} \exp\left(-\frac{t^2}{2}\right). \qquad (12)$$

We distinguish between the effect size of each match $m$ on the spread of COVID-19. For modeling the effect $\Delta R_{match,m}$, associated with public or private football screenings in the home country, we introduce one base effect $\Delta R_{match}^{mean}$ and a match specific offset $\Delta\alpha_m$ for a typical hierarchical modeling approach (dark orange box in Fig. 4). As

prior we assume that the base effect $\Delta R_{match}^{mean}$ is centered around zero, which means that in principle also a negative effect of the football matches can be inferred:

$$\Delta R_{match,m} = \alpha_{prior,m} \left(\Delta R_{match}^{mean} + \Delta\alpha_m\right) \qquad (13)$$

$$\Delta R_{match}^{mean} \sim \mathcal{N}(0,5) \qquad (14)$$

$$\Delta\alpha_m \sim \mathcal{N}(0,\sigma_\alpha) \qquad (15)$$

$$\sigma_\alpha \sim \text{HalfNormal}(5). \qquad (16)$$

**Table 5 | Maximal $\mathcal{R}$-hat values[51]**

| Country | Max. $\mathcal{R}$-hat of relevant variables | Max. $\mathcal{R}$-hat of all variables |
|---|---|---|
| England | 1.07 | 1.98 |
| The Czech Republic | 1.00 | 1.16 |
| Scotland | 1.01 | 1.10 |
| Spain | 1.05 | 2.24 |
| Italy | 1.01 | 1.10 |
| Slovakia | 1.00 | 1.15 |
| Germany | 1.01 | 1.42 |
| Austria | 1.00 | 1.15 |
| Belgium | 1.01 | 1.22 |
| France | 1.01 | 1.82 |
| Portugal | 1.00 | 1.14 |
| The Netherlands | 1.03 | 1.83 |

The convergence is good (≈1) for the relevant variables, which are the variables that encode the reproduction number.

$\alpha_{\text{prior},m}$ is the $m$-th element of the vector that encodes the prior expectation of the effect of a match on the reproduction number. If a country participated in a match, the entry is 1 and otherwise 0. The robustness of the results with respect to the hyperprior $\sigma_\alpha$ is explored in Supplementary Fig. S17.

For Supplementary Fig. S9, we expand the model by including the effect of infections happening in stadiums and in the vicinity of it as well as during travel towards the venue of the match. In detail, we add to the football related additive reproduction number (Eq. (11)) an additive effect $\Delta R_{\text{stadium},m}$:

$$\Delta R_{\text{football}}(t) = \sum_m (\Delta R_{\text{match},m} + \Delta R_{\text{stadium},m}) \cdot \delta(t_m - t). \tag{17}$$

Analogously to the gathering-related effect we apply the same hierarchy to the effect caused by hosting a match in the stadium – but change the prior of the day of the effect:

$$\Delta R_{\text{stadium},m} = \beta_{\text{prior},m}(\Delta R_{\text{stadium}}^{\text{mean}} + \Delta\beta_m) \tag{18}$$

$$\Delta R_{\text{stadium}}^{\text{mean}} \sim \mathcal{N}(0,5) \tag{19}$$

$$\Delta\beta_m \sim \mathcal{N}(0,\sigma_\beta) \tag{20}$$

$$\sigma_\beta \sim \text{HalfNormal}(5). \tag{21}$$

$\beta_{\text{prior},m}$ encodes whether or not a match was *hosted* by the respective country, i.e equates 1 if the match took place in the country and otherwise equates 0.

**Non-football-related reproduction number**

To account for effects not related to the football matches, e.g., non-pharmaceutical interventions, vaccinations, seasonality or variants, we introduce a slowly changing reproduction number $R_{\text{base}}(t)$, which is identical for both genders and should map all other not specifically modeled gender independent effects (left yellow box in Fig. 4):

$$R_{\text{base}}(t) = R_0 \exp\left(\sum_n \gamma_n(t)\right) \tag{22}$$

$$R_0 \sim \text{LogNormal}(\mu=1, \sigma=1) \tag{23}$$

This base reproduction number is modeled as a superposition of logistic change points $\gamma(t)$ every 10 days, which are parameterized by the transient length of the change points $l$, the date of the change point $d$ and the effect of the change point $\Delta\gamma_n$. The subscripts $n$ denotes the discrete enumeration of the change points:

$$\gamma_n(t) = \frac{1}{1 + e^{-4/l_n \cdot (t-d_n)}} \cdot \Delta\gamma_n \tag{24}$$

$$\Delta\gamma_n \sim \mathcal{N}(0,\sigma_{\Delta\gamma}) \,\forall n \tag{25}$$

$$\sigma_{\Delta\gamma} \sim \text{HalfCauchy}(0.5) \tag{26}$$

$$l_n = \log\left(1 + \exp(l_n^\dagger)\right) \tag{27}$$

$$l_n^\dagger \sim \mathcal{N}(4,1) \,\forall n \,(\text{unit is days}) \tag{28}$$

$$d_n = 27^{\text{th}} \text{ May } 2021 + 10 \cdot n + \Delta d_n \text{ for } n = 0, \ldots, 9 \tag{29}$$

$$\Delta d_n \sim \mathcal{N}(0,3.5) \,\forall n \,(\text{unit is days}). \tag{30}$$

The idea behind this parameterization is that $\Delta\gamma_n$ models the change of R-value, which occurs at times $d_n$. These changes are then summed in Eq. (24). Change points that have not occurred yet at time $t$ do not contribute in a significant way to the sum as the sigmoid function tends to zero for $t \ll d_n$. The robustness of the results regarding the spacing of the change-points $d_n$ is explored in Supplementary Fig. S14 and the robustness of the choice of the hyperprior $\sigma_{\Delta\gamma}$ is explored in Supplementary Fig. S17.

Similarly, to account for small changes in the gender imbalance, the noise on the ratio between infections in men and women is modeled by a slowly varying reproduction number (middle yellow box in Fig. 4), parameterized by series of change points every 10 days:

$$\Delta R_{\text{noise}}(t) = \Delta R_{0,\text{noise}} + \left(\sum_n \tilde{\gamma}_n(t)\right) \tag{31}$$

$$\Delta R_{0,\text{noise}} \sim \mathcal{N}(\mu=0, \sigma=0.1) \tag{32}$$

$$\tilde{\gamma}_n(t) = \frac{1}{1 + e^{-4/\tilde{l}_n \cdot (t-\tilde{d}_n)}} \cdot \Delta\tilde{\gamma}_n \tag{33}$$

$$\Delta\tilde{\gamma}_n \sim \mathcal{N}(0,\sigma_{\Delta\tilde{\gamma}}) \tag{34}$$

$$\sigma_{\Delta\tilde{\gamma}} \sim \text{HalfCauchy}(0.2) \tag{35}$$

$$\tilde{l}_n = \log\left(1 + \exp(\tilde{l}_n^\dagger)\right) \tag{36}$$

$$\tilde{l}_n^\dagger \sim \mathcal{N}(4,1) \,\forall n \,(\text{unit is days}) \tag{37}$$

$$\tilde{d}_n = 27^{\text{th}} \text{ May } 2021 + 10 \cdot n + \Delta\tilde{d}_n \text{ for } n = 0, \ldots, 9 \tag{38}$$

$$\Delta \tilde{d}_n \sim \mathcal{N}(0,3.5)\ \forall n \ (\text{unit is days}). \tag{39}$$

## Delay

Modeling the delay between the time of infection and the reporting of it is an important part of the model (blue boxes in Fig. 4); it allows for a precise identification of changes in the infection dynamics because of football matches and the reported cases. We split the delay into two different parts: First we convolved the number of newly infected people with a kernel, which delays the cases between 4 and 7 days. Second, to account for delays that occur because of the weekly structure (some people might delay getting tested until Monday if they have symptoms on Saturday or Sunday), we added a variable fraction that delays cases depending on the day of the week.

**Constant delay.** To account for the latent period and an eventual apparition of symptoms we apply a discrete convolution, a Gamma kernel, to the infected pool (right blue box in Fig. 4). The prior delay distribution $D$ is defined by incorporating knowledge about the country specific reporting structure: If the reported date corresponds to the moment of the sample collection (which is the case in England, Scotland and France) or if the reported date corresponds to the onset of symptoms (which is the case in the Netherlands), we assumed 4 days as the prior median of the delay between infection and case. If the reported date corresponds to the transmission of the case data to the authorities, we assumed 7 days as prior median of the delay. If we do not know what the published date corresponds to, we assumed a median $\bar{D}_{\text{country}}$ of 5 days, with a larger prior standard deviation $\sigma_{\log \bar{D}}$ (see Table 4):

$$C_g^{\dagger}(t) = \sum_{\tau=1}^{T} E_g(t-\tau) \cdot \text{Gamma}(\tau; \mu = D, \sigma = \sigma_D) \tag{40}$$

$$D = \log(D^{\dagger}) \tag{41}$$

$$D^{\dagger} \sim \mathcal{N}\left(\mu = \exp\left(\bar{D}_{\text{country}}\right), \sigma = \sigma_{\log \bar{D}}\right) \tag{42}$$

$$\sigma_D \sim \mathcal{N}\left(\mu = 0.2 \cdot \bar{D}_{\text{country}}, \sigma = 0.08 \cdot \bar{D}_{\text{country}}\right). \tag{43}$$

Here, Gamma represents the delay kernel. We obtain a delayed number of infected persons $C_g^{\dagger}$ by delaying the newly infected number of persons $I_g(t)$ of gender $g$ from Eq. (1). The robustness of the choice of the width of the delay kernel $\sigma_D$ is explored in Supplementary Fig. S17.

**Weekday-dependent delay.** Because of the different availability of testing resources during a week, we further delay a fraction of persons, depending on the day of the week (left blue box in Fig. 4). We model the fraction $\eta_t$ of delayed tests on a day $t$ in a recurrent fashion, meaning that if a certain fraction gets delayed on Saturday, these same individuals can still get delayed on Sunday (Eq. (44)). The fraction $\eta_t$ is drawn separately for each individual day. However, the prior is the same for certain days of the week $d$ (Eq. (45)): we assume that few tests get delayed on Tuesday, Wednesday, and Thursday, using a prior with mean 0.67% (Eq. (48)), whereas we assume that more tests might be delayed on Monday, Friday, Saturday and Sunday. Hence compared to $C_g^{\dagger}$, we obtain slightly more delayed numbers of cases $\hat{C}_g$, which now include a weekday-dependent delay:

$$\hat{C}_g(t) = (1-\eta_t) \cdot \left(C_g^{\dagger}(t) + \eta_{t-1}\hat{C}_g(t-1)\right) \text{ with } \hat{C}_g(0) = C_g^{\dagger}(0) \tag{44}$$

$$\eta_t \sim \text{Beta}\left(\alpha = \frac{r_d}{e}, \beta = \frac{1-r_d}{e}\right) \text{ with } d = \text{Monday},\dots,\text{Sunday} \tag{45}$$

$$r_d = \text{sigmoid}\left(r_d^{\dagger}\right) \tag{46}$$

$$r_d^{\dagger} = r_{\text{base},d}^{\dagger} + \Delta r_d^{\dagger} \tag{47}$$

$$r_{\text{base},d}^{\dagger} \sim \mathcal{N}(-5,1) \text{ for } d = \text{Tuesday, Wednesday, Thursday} \tag{48}$$

$$r_{\text{base},d}^{\dagger} \sim \mathcal{N}(-3,2) \text{ for } d = \text{Friday, Saturday, Sunday, Monday} \tag{49}$$

$$\Delta r_d^{\dagger} \sim \mathcal{N}(0,\sigma_r) \tag{50}$$

$$\sigma_r \sim \text{HalfNormal}(1) \tag{51}$$

$$e \sim \text{HalfCauchy}(0.2). \tag{52}$$

The parameter $r_d$ is defined such that it models the mean of the Beta distribution of Eq. (45), whereas $e$ models the scale of the Beta distribution. $r_d$ is then transformed to an unbounded space by the sigmoid $f(x) = \frac{1}{1+\exp(-x)}$ (Eq. (46)). This allows to define the hierarchical prior structure for the different weekdays. We chose the prior of $r_{\text{base},d}^{\dagger}$ for Tuesday, Wednesday, and Thursday such that only a small fraction of cases are delayed during the week. The chosen prior in Eq. (48) corresponds to a 2.5th and 97.5th percentile of $r_d$ of [0%; 5%]. For the other days (Friday, Saturday, Sunday, Monday), the chosen prior leaves a lot of freedom for inferring the delay. Equation (49) corresponds to a 2.5th and 97.5th percentile of $r_d$ of [0%; 72%]. The robustness of the other priors $\sigma_r$ and $e$ is explored in Supplementary Fig. S17.

## Likelihood

Next we want to define the goodness of fit of our model to the sample data. The likelihood of that is modeled by a Student's $t$-distribution, which allows for some outliers because of its heavier tails compared to a Normal distribution (green box in Fig. 4). The error of the Student's $t$-distribution is proportional to the square root of the number of cases, which corresponds to the scaling of the errors in a Poisson or Negative Binomial distribution:

$$C_g(t) \sim \text{StudentT}_{\nu=4}\left(\mu = \hat{C}_g(t), \sigma = \kappa \sqrt{\hat{C}_g(t)+1}\right) \tag{53}$$

$$\kappa \sim \text{HalfCauchy}(\sigma = 30). \tag{54}$$

Here $C_g(t)$ is the measured number of cases in the population of gender $g$ as reported by the respective health authorities, whereas $\hat{C}_g(t)$ is the modeled number of cases (Eq. (44)). The robustness of the prior $\kappa$ is explored in Supplementary Fig. S17.

## Average effect across countries

In order to calculate the mean effect size across countries (Fig. 1b, c), we average the individual effects of each country. To be consistent in our approach, we build an hierarchical Bayesian model accounting for the individual uncertainties of each country estimated from the width of the posterior distributions. As effect size, we use the fraction of primary cases associated with football matches during the championship. Then our estimated mean effect size $\hat{I}_g$ across all countries $c$ (except the Netherlands) for the gender $g$ is inferred with the following

model:

$$\hat{I}_g \sim \text{Normal}(\mu = 0, \sigma = 2) \quad \text{with } g = \{\text{male, female}\} \tag{55}$$

$$\tau_g \sim \text{HalfCauchy}(\beta = 10) \tag{56}$$

$$I_{c,g}^{\dagger} \sim \text{Normal}(\mu = \hat{I}_g, \sigma = \tau_g) \tag{57}$$

$$\hat{\sigma}_{c,g} \sim \text{HalfCauchy}(\beta = 10) \tag{58}$$

$$I_{s,c,g} \sim \text{StudentT}_{\nu = 4}\left(\mu = I_{c,g}^{\dagger}, \sigma = \hat{\sigma}_{c,g}\right). \tag{59}$$

The estimated effect size of each country (the fraction of primary cases) is denoted by $I_{c,g}^{\dagger}$ and the effect size of individual samples $s$ from the posterior of the main model is denoted by $I_{s,c,g}$.

We applied a similar hierarchical model but without gender dimensions and with slightly different priors to calculate the average mean match effect $\Delta R_{\text{match}}^{\text{mean}}$ (Fig. 1a). Here by reusing the same notation:

$$\hat{I} \sim \text{Normal}(\mu = 0, \sigma = 10) \tag{60}$$

$$\tau \sim \text{HalfCauchy}(\beta = 10) \tag{61}$$

$$I_c^{\dagger} \sim \text{Normal}(\mu = \hat{I}, \sigma = \tau) \tag{62}$$

$$\hat{\sigma}_c \sim \text{HalfCauchy}(\beta = 10) \tag{63}$$

$$\Delta R_{\text{match},c,s}^{\text{mean}} \sim \text{StudentT}_{\nu = 4}(\mu = I_c^{\dagger}, \sigma = \hat{\sigma}_c), \tag{64}$$

where $\Delta R_{\text{match},c,s}^{\text{mean}}$ are the posterior samples from the main model runs of the $\Delta R_{\text{match}}^{\text{mean}}$ variable.

## Calculating the primary and subsequent cases

We compute the number of primary football related infected $I_{\text{primary},g}(t)$ as the number of infections happening at football related gathering. The percentage of primary cases $f_g$ is then computed by dividing by the total number of infected $I_g(t)$.

$$I_{\text{primary},g}(t) = \frac{S(t)R_{\text{football}}(t)}{N} \sum_{g'} I_{g'}(t)\mathbf{C}_{\text{football},g',g} \tag{65}$$

$$f_g = \sum_t \frac{I_{\text{primary},g}(t)}{I_g(t)} \quad t \in [\text{11th June, 31st July}] \tag{66}$$

To obtain the subsequent infected $I_{\text{subsequent},g}(t)$, we subtract infected obtained from a hypothetical scenario without football games $I_{\text{none},g}(t)$ from the total number of infected.

$$I_{\text{subsequent},g} = I_g(t) - I_{\text{primary},g}(t) - I_{\text{none},g}(t) \tag{67}$$

Specific, we consider a counterfactual scenario, where we sample from our model leaving all inferred parameters the same expect for the football related reproduction number $R_{\text{football},g}(t)$, which we set to zero.

## Sampling

The sampling was done using PyMC3[49]. We use a NUTS sampler[50], which is a Hamiltonian Monte-Carlo sampler. As random initialization often leads to some chains getting stuck in local minima, we run 32

chains for 500 initialization steps and chose the 8 chains with the highest unnormalized posterior to continue tuning and sampling. We then let these chains tune for additional 2000 steps and draw 4000 samples. The maximum tree depth was set to 12.

The quality of the mixing was tested with the $\mathcal{R}$-hat measure[51] (Table 5). The $\mathcal{R}$-hat value measures how well chains with different starting values mix; optimal are values near one. We measured twice: (1) for all variables and (2) for the subset of variables encoding the reproduction number. Variables modeling the reproduction number are the central part of our model (lower half of Fig. 4). As such, we are satisfied if the $\mathcal{R}$-hat values is sufficiently good for these variables, which it is ($\leq$1.07). The high $\mathcal{R}$-hat when calculated over all variables is mostly due to the weekday-dependent delay, which we assume is not central to the results we are interested in.

## Robustness tests

In the base model for each country, we only consider the matches in which the respective country participated. It is reasonable to ask whether the matches of foreign countries occurring in local stadium have an effect on the case numbers, caused by transmission in and around the stadium and related travel. To investigate this question we ran a model with an additional parameter (in-country effect) associating the case numbers to the in-country matches (Eq. (17)). In some countries the in-country effect parameter and the original fan gathering effect are covariant, as a large number of matches are played by the country at home, whereas in other countries the additional parameter had no significant effect (Supplementary Fig. S9).

We checked that the inferred fractions of football related cases are robust against changes in the priors of the width $\sigma_D$ of the delay parameter $D$ (see Supplementary Fig. S13) and the intervals of change points of $R_{\text{base}}$ (Supplementary Fig. S14). The results are also, to a very large degree, robust against a more uninformative prior on the fraction of female participants in the fan activities dominating the additional transmission $\omega_{\text{gender}}$ (Supplementary Fig. S15). To reduce $CO_2$ emissions, we performed fewer runs for these robustness tests: We only ran the models for which the original posterior distributions might indicate that one could find a significant effect. Each country required eight cores for about 10 days to finish sampling.

In order to further test the robustness of the association between individual matches and infections, we varied the dates of the matches, i.e., shifted them forward and backward in time. The results for the twelve countries under investigation are shown in Supplementary Figs. S10 and S12. In the countries where sensitivity to a championship-related case surge exists, a stable association is obtained for shifts by up to 2 days. As shown for the examples of England and Scotland in Fig. S19, such a shift is compensated by the model by a complementary adjustment of the delay parameter $D$. For larger shifts, the model might associate other matches to the increase of cases, as matches took place approximately every 4 days.

## Correlations

In order to calculate the correlation between the effect size and various explainable variables (Fig. 3 and Supplementary Figs. S4 and S6), we built a Bayesian regression model, using the previously computed posterior samples from the individual runs of each country. Let us denote the previously computed cumulative primary and subsequent cases related to the Euro 2020 by $Y_{s,c}$, for every sample $s$ and analyzed country $c$, and the explainable variable from auxiliary data by $X_c$. We used a simple linear model to check for pairwise correlation between $Y_{s,c}$ and $X_c$:

$$\hat{Y}_c = \beta_0 + \beta_1 \hat{X}_c \tag{68}$$

$$\beta_0 \sim \text{Normal}(\mu = 0, \sigma = 10000) \tag{69}$$

$$\beta_1 \sim \text{Normal}(\mu = 0, \sigma = 100000) \tag{70}$$

We used every sample $s$ obtained from the main analysis to incorporate uncertainties on the variable $Y_c$ from our prior results. The auxiliary data $X_c$ might also have errors $\epsilon_c$, which we model using a Normal distribution. Additionally, we allow our estimate for the effect size $\hat{Y}_c$ to have an error for each country $c$ in a typical hierarchical manner and choose uninformative priors for the scale hyper-parameter $\tau$. As prior we considered 10k a reasonable choice for the $\beta$ parameter as our data $X_c$ is normally in a range multiple magnitudes smaller:

$$\hat{X}_c \sim \text{Normal}(\mu = X_c, \sigma = \epsilon_c) \; \forall c \tag{71}$$

$$\tau \sim \text{HalfCauchy}(\beta = 10000) \tag{72}$$

$$Y_c^\dagger \sim \text{Normal}(\mu = \hat{Y}_c, \sigma = \tau) \; \forall c. \tag{73}$$

Again using uninformative priors for the error, the likelihood to obtain our results given the individual country effect size estimate $Y_c^\dagger$ from the hierarchical linear model is

$$Y_{s,c} \sim \text{StudentT}_{\nu=4}\left(\mu = Y_c^\dagger, \sigma = \hat{\sigma}_c\right) \; with \tag{74}$$

$$\hat{\sigma}_c \sim \text{HalfCauchy}(\beta = 10000). \tag{75}$$

Therefore, our regression model includes the "measurement error" $\hat{\sigma}_c$, which models the heteroscadistic effect size of every country, and an additional model error $\tau$ which models the homoscedastic deviations of the country effect sizes from the linear model. In the plots, we plot the regression line $\hat{Y}_c$ with its shaded 95% CI, and data points $(\hat{X}_c, Y_c^\dagger)$ where the whiskers correspond to the one standard deviation, modeled here by $\epsilon_c$ and $\hat{\sigma}_c$.

The coefficient of determination, $R^2$, is calculated following the procedure suggested by Gelman and colleagues[52]. Their $R^2$ measure is intended for Bayesian regression models as it notably uses the expected data variance given the model instead of the observed data variance. For our model, it is defined as

$$R^2 = \frac{\text{Explained variance}}{\text{Residual variance} + \text{Explained variance}} = \frac{\frac{1}{n_c - 1}\sum_c \hat{Y}_c^2}{\tau^2 + \frac{1}{n_c - 1}\sum_c \hat{Y}_c^2}, \tag{76}$$

where $n_c$ is the number of countries. With this formula, one obtains the posterior distribution of $R^2$ by evaluating it for every sample.

As auxiliary data, we used:

1. *Mobility data*: We use the mobility index $m_{c,t}$ provided by the "Google COVID-19 Community Mobility Reports"[53] for each country $c$ at day $t$ during the Euro 2020 ($t \in$ [June 11 2021, July 11 2021]), where $N$ denotes the number of days in the interval. The error is the standard deviation of the mean:

$$X_c = \frac{1}{N}\sum_t m_{c,t} \tag{77}$$

$$\epsilon_c = \sqrt{\frac{1}{N^2}\sum_t (m_{c,t} - X_c)^2} \tag{78}$$

2. *Reproduction number*: We use the base reproduction number $R_{\text{pre},c}$ for each country $c$ as inferred from our model 2 weeks prior

to the Euro 2020 ($t \in$ [May 28 2021, June 11 2021]).

$$X_c = \frac{1}{N}\sum_t R_{\text{pre},c}(t) \tag{79}$$

$$\epsilon_c = \sqrt{\frac{1}{N}\sum_t (R_{\text{pre},c}(t) - X_c)^2} \tag{80}$$

3. *Cumulative reported cases*: From the daily reported cases $C(t)$ two weeks prior to the Euro 2020 ($t \in$ [May 28 2021, June 11 2021]), we computed the cumulative reported cases normalized by the number of inhabitants $p_c$ in each country $c$. Note: We also used reported cases without gender assignment here.

$$X_c = \frac{\sum_t C(t)}{p_c} \tag{81}$$

$$\epsilon_c = \vec{0} \tag{82}$$

4. *Potential for COVID-19 spread*: As for the cumulative cases we used the daily reported cases $C(t)$ two weeks prior to the Euro 2020 ($t \in$ [May 28 2021, June 11 2021]), and we computed the cumulative reported cases normalized by the number of inhabitants $p_c$ in each country $c$. Furthermore, we used the base reproduction number $R_{\text{base}}(t)$ 2 weeks prior to the Euro 2020, as well as the duration of a country participating in the championship $T_c$ (Table S5) to compute the potential for spread:

$$N_0 = \frac{\sum_t C(t)}{p_c} \tag{83}$$

$$X_c = N_0 \cdot \frac{\sum_t R_{\text{pre},c}^{T_c/4}(t)}{N} \tag{84}$$

$$\epsilon_c = \vec{0} \tag{85}$$

5. *Proxy for popularity*: To represent popularity of the Euro 2020 in country $c$, we used the union of the number of matches played by each country $n_{\text{match},c}$ and the number of matches hosted by each country $n_{\text{hosted},c}$ (Table S5). By "union" we mean the sum without the overlap, i.e., we take the sum of these numbers and subtract the number of home matches $n_{\text{home},c}$

$$X_c = n_{\text{match},c} + n_{\text{hosted},c} - n_{\text{home},c} \tag{86}$$

$$\epsilon_c = \vec{0} \tag{87}$$

## Reporting summary

Further information on research design is available in the Nature Portfolio Reporting Summary linked to this article.

## Data availability

The data from our model runs, i.e., from the sampling is available on G-node https://gin.g-node.org/semohr/covid19_soccer_data. The daily case numbers stratified by age and gender were acquired from the local health authorities (see also Supplementary section S1 from the following sources: Robert Koch Institut, Germany; Santé publique, France; National Health Service, England; Österreichische Agentur für Gesundheit und Ernährungssicherheit GmbH, Austria; Sciensano, BelgiumMinisterstvo zdravotnictví, Czech Republic; National Institute for Public Health and the Environment, The Netherlands; and COVerAGE-DB.

## Code availability

All code to reproduce the analysis and figures shown in the manuscript as well as in the Supplementary Information is available online on GitHub https://github.com/Priesemann-Group/covid19_soccer or via https://doi.org/10.5281/zenodo.7386313[54].

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

## Acknowledgements

We thank the Priesemann group for exciting discussions and for their valuable input. We also thank Arne Gottwald and Cornelius Grunwald for their continuous input during the conceptual and finalizing stage of the manuscript. We thank Piklu Mallick for proofreading. All authors received support from the Max-Planck-Society. J.D. and S.B.M. received funding from the "Netzwerk Universitätsmedizin" (NUM) project egePan (01KX2021). S.B.M. received funding from the "Infrastructure for exchange of research data and software" (crc1456-inf) project. S.C. and P.D. received funding by the German Federal Ministry for Education and Research for the RESPINOW project (031L0298) and ENI for the infoXpand project (031L0300A). V.P. was supported by the Deutsche Forschungsgemeinschaft (DFG, German Research Foundation) under Germany's Excellence Strategy - EXC 2067/1-390729940. This work was partly performed in the framework of the PUNCH4NFDI consortium supported by DFG fund "NFDI 39/1", Germany.

## Author contributions

Conceptualization: V.P., S.B.M., J.D., S.C., P.D. Methodology: S.B.M., J.D., V.P., P.B., O.S. Software: S.B.M., J.D. Validation: S.B.M., J.D. Formal analysis: S.B.M., J.D. Investigation: S.B.M., J.D., O.S., P.B. Data curation: S.B.M., J.D., P.D., O.S. Writing—original draft: all. Writing—review and editing: all. Visualization: S.B.M., J.D., P.D., S.C., E.I. Supervision: V.P., P.B., O.S., J.D. Funding acquisition: V.P., P.B., O.S.

## Funding

## Competing interests

V.P. is a member of the ExpertInnenrat of the German federal government on COVID and is also advising other governmental and non-governmental entities. The other authors declare no competing interests.
