## [Peer Review File · Nature Communications]

Impact of the Euro 2020 championship on the spread of COVID-19REVIEWER COMMENTS

Reviewer #1 (Remarks to the Author):

This article presents a Bayesian model to appreciate the impact of isolated events, here "football matches" on the dynamics of transmission of a communicable disease, here "COVID-19". A model is built where extra transmission surrounds football matches, implicating 2 teams, and the country hosting the match. Sex imbalance in cases helps estimating increases in transmission.

1-The authors claim (line 27) that euro 2020 is a "randomized trial" between countries and that this help them making causal conclusions.

I do not see how their analysis uses this fact, nor how it could inform causality.

In randomized trials, randomization generally ensure that all units are exchangeable between treatment arms,

so that effect is obtained by simple differences.

Here a model is fit without any reference to the random nature of EURO2020. The justification is not clear in the introduction

and furthermore this aspect is not mentioned again in the discussion nor to justify causal interpretation.

If the authors believe that the randomization can really help in their study, this should be argued much more convincingly.

Otherwise, it should be removed.

2-The authors make a strong assumption that part of sex imbalance in cases is due to EURO2020; and estimate the effect of football matches conditional on this assumption.

The model assumes that effects are only seen in either countries having a match, or in hosting countries.

It could be that the effect of matches are seen in every countries for every match.

Did the author try to fit such a model and would it be feasible ?

3-In Fig 3A, it's not clear how R_{base} is defined. It is varying with time. Furthermore, given how the number of secondary cases is computed in the model, which is a function of $R_{base}(t)$ and generation interval, it seems obvious that it should scale with $R^{(T/4)}$. Could it be shown that this conclusion is not foregone?

4-Is ω_{gender} really a measure of "as likely to attend football related"? It seems more related to the composition of the population with 33% women and that women are 50% as likely to attend.

5-I'm not convinced that the lack of convergence of the daily parameters as indicated by the H statistics is not relevant.

Since the authors build on the precise timing of events to infer parameters, this may on the contrary have a strong effect.

It is customary to report the traces of estimated parameters to illustrate convergence; this could be done here in the supplementary material.

6-Is the unspecific contact matrix for sex imbalance really with a -1 ? is this for centering? a few words may be of use.

7-The authors report the Oxford tracker. Could they find a relationship between the stringency of measures and the effect of matches?

maybe a correlation between stringency at the time of the match and estimated match effect?

Mobility was discussed in this respect, but I couldn't find summary measures.

Reviewer #2 (Remarks to the Author):

Main Comments:

1) A number of the conclusions appear to rely heavily on the fact that England and Scotland saw large gender imbalances in Covid-19 cases, with other countries contributing substantially less to the conclusions. This is particularly true in Figure 3A, where the points corresponding to England and Scotland have very high leverage and so will dominate the slope of the line of best fit. To what extent therefore are these results internationally applicable - is it possible that cultural differences in the United Kingdom are uniquely responsible for football matches causing a gender imbalance in case numbers? What would the results (including Figure 3A) look like if the UK (i.e. England and Scotland) were excluded? What issues arise from treating England and Scotland as independent, despite them being part of the same country, subject to the same national-level measures?

2) The authors' justification for removing the Netherlands from the analysis is reasonable, but it is concerning to see the extent to which there was a gender imbalance in cases towards women. Under the model used in this paper, what is the probability that such a large deviation occurred due to random noise alone? Moreover, if this probability is small, is it possible that there is insufficient noise assumed in the model, and hence that some of the significance of the results in countries which saw a gender imbalance towards men has been over-stated?

3) In equations (22) and (31), it appears that the sums are over all n - assumedly the sum should only be over those n such that the changepoint associated with n has already occurred?

4) Prior distributions are chosen throughout (e.g. equations (9), (16), (21)...) without justification. To what extent are the results dependent on these choices? It is important that the sensitivity of the conclusions is explored sufficiently to inform readers (and indeed referees) of their strength. This should not be avoided citing "environmental reasons".

5) If environmental concerns, as cited in Figures S9 and S10, are valid, then can the authors determine computational efficiencies that would make the investigation more feasible without unreasonable environmental costs?

6) Would it be possible to run the same analysis, but to initialise the model significantly before the start of the championship? This would provide a good examination of the interaction between the base and noise terms.

7) Why are the p-values given for one-sided, rather than two-sided, tests? [For example, in Figure S6.] Indeed Figure S6(C) looks like the two-sided p-value should be considerably greater than 0.06. Also why is the central line so near the top of the shaded area? This would appear to be an error which makes me concerned about all of the graphs with such shading.

8) Do the linear regression models plotted in Figure 3 (and elsewhere) allow for heteroscedasticity? If not, then this should be allowed for given the considerable variability in uncertainty associated with the plotted estimates. The methods section should make 100% clear what how linear regressions have been estimated and what is plotted.

Editorial Comments:

1) Be consistent with decimal / comma notation (e.g. 10.000 is used in the abstract to mean what is given as 10000 elsewhere)

2) Line 29: There is potentially some link between pandemic state and team progression (e.g. Billy Gilmour was forced to isolate for Scotland).

3) Line 166: Should be "FIFA World Cup" instead of "World Championship"

4) Table S2, row 5: Typo - you have $R_{\{b\}ase}$ instead of $R_{\{base\}}$

5) Line 389: $\alpha_{\{prior,m\}}$ is acting as a function of m rather than a matrix (also in Table S1)

6) Why is $\beta_{\{prior,m\}}$ not listed in Table S1?

7) Inconsistency throughout between "The Czech Republic" and "Czechia"

8) In Figure S7, the terms "Quarter-finals" and "Semifinal" are fine, but there is only one "Final" - it is not "Finals". The terms are not "quarter finale" [Fig S19] or "finale" [Fig S15]. Further, the

paper should not switch between "final match" and "finals". The text (line 98) is confusing when it refers to "final matches" since there is only one "final match".

9) Line 442: Should be "eq. (44)" – i.e. the brackets are needed.

10) Table S10: "Time in championship" should really be time between first and last match, shouldn't it?

11) In all cases the figure captions need to clearly define the shading as well as any lines that have been plotted. For example, it would appear that Fig S5 shows linear regression model estimates and 95% confidence intervals, but this is not stated.

Reviewer #3 (Remarks to the Author):

This study aims to quantify the impact of the UEFA Euro 2020 Football Championship on the spread of COVID-19 among 12 countries to influence public health policy.

This is an interesting paper that exemplifies the importance of public health policies regarding large-scale sporting events. I found one major limitation in the estimation of the number of deaths associated with the analyzed events and a set of other relatively easily addressable points.

- Line 37, "disease transmission rates" -> "infection transmission rates". The disease cannot be transmitted; the infection (or the pathogen) is transmitted.

- Line 47. "Basic" should be "base" (according to the nomenclature used in the rest of the manuscript).

- Line 64. Primary cases are defined as "infections occurring at gatherings on match days." How are these primary cases identified? And how do you differentiate 1) between primary and subsequent cases and 2) cases that occur from different matches?

- In lines 66-67, you mention "We included all subsequent until July 31...". Were subsequent cases for all participating countries analyzed until July 31 or was that only for the countries involved in the final match? If yes, how do you justify that countries participating only in early matches are still contributing to subsequent COVID-19 cases long after the matches? If all countries were not included until July 31, were subsequent cases two weeks after the country's final match included in the analysis?

- Line 78. First, it is SARS-CoV-2 infections and not COVID-19 infections. Second, these are "reported SARS-CoV-2 infections", which are large underestimations of the true number of infections. Please rephrase and add a comment on this in the Discussion.

- Line 79. First, that is a "case fatality ratio", not a rate. Rates are expressed in time^{-1} , while you are using that as a ratio instead. Second, exactly as there is a gender imbalance in the population affected by Euro 2020, there very likely is an age imbalance as well. Specifically, we expect that population to be much younger than the general population of the country. As such, for a disease like COVID-19 where the fatality is much higher in the elderly, applying an age-independent case fatality ratio provides hardly credible results. I strongly encourage the authors to either to rely on age-dependent estimates of the case fatality ratio or to entirely drop the estimates of the number of deaths.

- Connected to the previous point, it is possible that the case reporting rate has temporarily increased right after each match. This should be discussed as a study limitation.

- Line 122. A generation interval of 4 days appears to be very short. That could be a reasonable estimate for a Chinese setting with very isolation policies in dedicated facilities, but rather short for a European context with very loose household isolation policies. 6 days would be a more sensible choice (see for instance Manica et al, Estimation of the incubation period and generation time of SARS-CoV-2 Alpha and Delta variants from contact tracing data, medrxiv).

- Lines 145-147 and 148-150. These sentences are speculative. It might well be the case that such

events should be entirely banned during certain epidemic phases and/or mass gatherings avoided altogether. Moreover, the authorities "should" not do anything based on a manuscript. Each authority should make the decision based on its specific targets and priorities (which may not be aligned with those considered in this manuscript).

- Line 150-151. I agree with this point, but it is phrased rather badly. The incubation period has a wide distribution, and its mean is not representative of the whole phenomenon. Moreover, not only the mean of the incubation period but also the mean of the generation time is in line with the interval between matches.

- In Figures 2 and S4, base cases are named "independent cases" in the figure, but the captions and main text all refer to them as "base cases". I suggest keeping these labels consistent throughout the paper and figures.

- In general, there is quite a bit of confusion between cases and infections that the authors appear to be used interchangeably, while they are two clearly defined and different epidemiological concepts. Please carefully revise the wording throughout the manuscript.

October 21, 2022

Revision of our manuscript to *Nature Communications*

Dear reviewers,

thank you very much for the helpful comments!

Following the suggestions, we added additional analyses and clarifications to the manuscript and the Supplementary Information. To summarize the main points:

- We ran additional robustness checks on a number of priors (Supplementary Fig. S18) and the generation interval (Supplementary Fig. S17)
- For the already existing robustness checks, we added runs of the remaining countries (Supplementary Fig. S13–18)
- We added a consistency check by offsetting the matches by ± 30 , ± 35 , and ± 40 days to show that during time-periods where we do not expect an effect, our model does not infer an effect (Supplementary Fig. S12).
- We added plots to illustrate the sampling of our Markov Chain Monte Carlo chains and to show that the chains are well mixed in for our variables of interest, even if individual degenerate parameters of our model do not showcase good mixing (Supplementary Fig. S38–S49).
- We added a new figure showing an analysis of the mean and standard deviation of the gender imbalance before and during the Euro 2020 (Supplementary Fig. S22). Here we clearly show that during the Euro 2020 the mean and variance increased on average.

We once again thank you for your valuable input and are looking forward to your reply,

Viola Priesemann and Philip Bechtle
(on behalf of all authors)

Reviewer 1

This article presents a Bayesian model to appreciate the impact of isolated events, here football matches on the dynamics of transmission of a communicable disease, here COVID-19. A model is built where extra transmission surrounds football matches, implicating 2 teams, and the country hosting the match. Sex imbalance in cases helps estimating increases in transmission.

We thank you for your helpful comments and suggestions, which led us to expand our manuscript, especially the controls and consistency checks. Below we address them point by point.

1 The authors claim (line 27) that euro 2020 is a randomized trial between countries and that this help them making causal conclusions. I do not see how their analysis uses this fact, nor how it could inform causality. In randomized trials, randomization generally ensure that all units are exchangeable between treatment arms, so that effect is obtained by simple differences. Here a model is fit without any reference to the random nature of EURO2020. The justification is not clear in the introduction and furthermore this aspect is not mentioned again in the discussion nor to justify causal interpretation. If the authors believe that the randomization can really help in their study, this should be argued much more convincingly. Otherwise, it should be removed.

Thank you for pointing this out. We agree that our study is not a “controlled randomized trial” in the strict sense. However, our study benefits from a specific kind of randomization that enables us to at least approximately extract a causal effect. Other typical studies that infer the effect of social gatherings by quantifying, e.g. non-pharmaceutical interventions (NPIs), have the bias that the NPIs are typically correlated with the incidence (e.g., with increasing incidence, the NPIs become stronger as well). In our case the time points of the matches do not depend on the incidence in the countries or their change. Moreover, a team’s success in a match is, in principle, independent of the incidence in the given country (see also our reply to comment **E2** of Reviewer 2). Thus overall, randomization in that sense is present and, thereby, the source of bias present in said other studies is eliminated.

To enable us to refer to this benefit of (approximate) randomization, we changed the phrasing in our text to call it a “randomized study” to distinguish it from controlled randomized trials, and we make explicit what we mean by it. Moreover, we removed the reference to Banerjee et al. 2016. The introduction paragraph (lines 27–33) reads now:

“ Two facts make the Euro 2020 especially suitable for the quantification. First, the Euro 2020 resembles a randomized study across countries: The time-points of the matches in a country do not depend on the state of the pandemic in that country and how far a team advances in the championship has a random component as well [20]. This independence between the time-points of the match and the COVID-19 incidence allows quantifying the effect of football-related social gatherings without classical biasing effects. This is advantageous compared to classical inference studies quantifying the impact of non-pharmaceutical interventions (NPIs) on COVID-19 where implementing NPIs is a typical reaction to growing case numbers [Dehning2020, Brauner2020, Sharma2021]. ”

2 The authors make a strong assumption that part of sex imbalance in cases is due to EURO2020; and estimate the effect of football matches conditional on this assumption. The model assumes that effects are only seen in either countries having a match, or in hosting countries. It could be that the effect of matches are seen in every countries for every match. Did the author try to fit such a model and would it be feasible?

Thank you for the interesting idea. However, we would not be able to successfully fit such a model because there would be too many matches (51) for the duration the tournament (30 days) and the number of days with matches (22) to have statistical power. In a slightly earlier version of the model, we concentrated on the potentially strongest effect, namely testing whether we see the effect of the final and semi-final of England in the Scottish case numbers, but there was no significant effect that our model could find (see below in gray):

To test for the potential effect on gender-imbalance, we compare the time before and during the tournament. One expects a larger gender imbalance and a larger variance

of the gender imbalance during during the tournament, compared to the time before. Specifically, we estimated the mean and the standard deviation of the gender imbalance during the 30 days of the tournament (plus the 5 days after) and for the 35 days before the tournament; we see on average a clear difference: Both, the mean and the variance of gender imbalance, were typically much larger during the tournament, indicating that the matches induced strong fluctuations in gender imbalance. This is shown in the figure that we now added to the manuscript (Supplementary Fig. S22, right column).

Regarding the remark “The authors make a strong assumption that part of sex imbalance in cases is due to EURO2020”. The main assumption we are making is that we *allow* differing reproduction numbers on the day of the matches: We find similar effect sizes even if we choose an alternative prior assumption, namely that the female and male fraction are equal (Supplementary Figure S16, purple histograms, gray is the prior). Even for this case, the posterior distribution of the female participation converge for the three significant countries to median values between 20% – 45%. We now make this clear in lines 200–202:

“ Furthermore, when using wider prior ranges for the gender imbalance, football-related COVID-19 cases remain unchanged but the uncertainty increases (Supplementary Fig. S16), thus validating our choice. **Even for the case of prior symmetric gender imbalance assumptions, the posterior distribution of the female participation converge for the three most significant countries to median values between 20% – 45%. ”**

3 In Fig 3A, its not clear how R_{base} is defined. It is varying with time. Furthermore, given how the number of secondary cases is computed in the model, which is a function of $R_{base}(t)$ and generation interval, it seems obvious that it should scale with $R^{(T/4)}$. Could it be shown that this conclusion is not foregone?

Thank you for this remark. We agree that the notation was a bit confusing. We now specifically named the “ R_{base} directly before the championship” R_{pre} and made sure to clarify that is is the reproduction number prior to the championship lines 125-127:

“ From theory, we expect the absolute number of infections generated by Euro 2020 matches to depend non-linearly on a country’s base incidence N_0 , which determines the probability to meet an infected person, and on the effective reproduction number **prior to the championship R_{pre} , as a gauge for the underlying infection dynamics generating the subsequent cases, which determines how strongly an additional infection spreads in the population. ”**

And also in lines 136-140:

“ Altogether, our data suggest that a favorable pandemic situation (low R_{pre} and low N_0) before the gatherings, and low R_{base} during the period of gatherings jointly minimize the impact of the Euro 2020 on community contagion. A prerequisite for this is that the known preventive measures, such as reducing group size, imposing preventive measures, and minimizing the number of encounters remain encouraged. ”

We agree that some degree of correlation with R_{pre} is not surprising. However, it is only observable as long as not the primary infections at the gatherings but the chains of subsequent cases dominate the overall impact of the championship. The fact that the quantitative effect of the pandemic situation before the championship (R_{pre}) has a clear impact is an important finding for preventive mitigation practices.

4 Is ω_{gender} really a measure of as likely to attend football related? It seems more related to the composition of the population with 33% women and that women are 50% as likely to attend.

Thank you for pointing this out. Our description of the normalization was not precise. We expanded the definition in Supplementary Table S2, and changed line 412ff to make it more clear:

“ ω_{gender} | The fraction of female participation in football related gatherings compared to the total participation ”

“ C_{match} describes the contact behavior in the context of the Euro 2020 football matches (right purple box in Supplementary Fig. S1). Here, we assume as a prior that the female participation in football-related gatherings accounts for $\simeq 33\%$ (95% percentiles [18%,51%] of the total participation. ”

5 Im not convinced that the lack of convergence of the daily parameters as indicated by the H statistics is not relevant. Since the authors build on the precise timing of events to infer parameters, this may on the contrary have a strong effect. It is customary to report the traces of estimated parameters to illustrate convergence; this could be done here in the supplementary material.

Thank you for raising this point. We now report the traces to illustrate convergence (see Supplementary Figures S38–S49). We show that for the parameters of interest the lack of convergence by the \mathcal{R} -hat statistics is not relevant. The remaining non-convergent chains are due to interchanging degenerate solutions in auxiliary parameters, as exemplified in Supplementary Figure S38. We observe good mixing of the chains for the parameters of interest. Even if some chains are more biased in some

parameters, the effect on our parameters of interest is small. The lack of convergence of daily parameters are due to some degeneracy in the parameters: We could in principle create a model without daily reporting delay, which would have no degenerate solutions for auxiliary parameters but would also be less realistic.

6 Is the unspecific contact matrix for sex imbalance really with a -1 ? is this for centering? a few words may be of use.

Thank you for the feedback. We agree that the paragraph explaining C_{noise} was lacking. We included a sentence explaining the centering (lines 420–421):

“ C_{noise} describes the effect of an additional noise term, which changes gender balance without being related to football matches (middle purple box in Supplementary Fig. S1). For simplicity, it is implemented as

$$C_{\text{noise}} = \begin{pmatrix} 1 & 0 \\ 0 & -1 \end{pmatrix}, \quad (1)$$

whereby we center the diagonal elements, such that the cases introduced by the noise term sum up to zero, i.e. $\sum_{i,j} R_{\text{noise}} \cdot C_{\text{noise},i,j} = 0$. ”

7 The authors report the Oxford tracker. Could they find a relationship between the stringency of measures and the effect of matches? maybe a correlation between stringency at the time of the match and estimated match effect? Mobility was discussed in this respect, but I couldnt find summary measures.

This is indeed an interesting question. We have already looked into the stringency measures before submission of the manuscript, and have included it in the analysis in the manuscript now (see Supplementary Figure S6) . We made this clear by including the following sentence in lines 146–147:

“ Moreover, we found no relationship between the effect size and the Oxford governmental response tracker [Hale2021] (Supplementary Fig. S6). ”

This can also be seen by comparing panels A and C in the Supplementary Figures S25–S37.

Reviewer 2

1 A number of the conclusions appear to rely heavily on the fact that England and Scotland saw large gender imbalances in Covid-19 cases, with other countries contributing substantially less to the conclusions. This is particularly true in Figure 3A, where the points corresponding to England and Scotland have very high leverage and so will dominate the slope of the line of best fit. To what extent therefore are these results internationally applicable - is it possible that cultural differences in the United Kingdom are uniquely responsible for football matches causing a gender imbalance in case numbers? What would the results (including Figure 3A) look like if the UK (i.e. England and Scotland) were excluded? What issues arise from treating England and Scotland as independent, despite them being part of the same country, subject to the same national-level measures?

Indeed we assume that there are unknown cultural aspects at play in each country. Therefore, our model makes no assumption about the cultural background of the strongly varying gender imbalances within countries. We do not dare speculate in the paper as to which specific cultural effects cause the observed strongly differing gender imbalances. E.g. we observe potentially different gender imbalances between England (0.32 [0.28,0.38]) and Czech Republic (0.41 [0.29,0.51]), showing that the model can indeed attribute soccer related cases for different ranges of observed gender imbalances, even if this range includes 50% at the 95% confidence level. Since the model has this freedom independently in each country, we believe that separating the countries by observed effect size post-hoc is not necessarily statistically representative.

Nonetheless, it is of course interesting to look at such a split model. Fig. 3A specifically is now shown without the UK in Supplementary Fig. S8.

As expected when removing the most significant data points, the overall significance is reduced. However, the regression parameters are consistent between all data points (1.62 [1.0, 2.26]) and the result without the UK (0.76 [-1.46, 3.04]). We now mention this in the document in lines 130-133:

“ The strong significance of this correlation relies mainly on England and Scotland. However, the observed trend in an analysis without these two countries, while not significant at the 95% confidence level, is consistent with the findings including all countries. This is shown in Supplementary Fig. S8. ”

We also reran our analysis not assuming England and Scotland as independent: We added the case numbers of both constituent countries and combined their matches on this run. We found overall similar results (Supplementary Fig. S19 and S37).

2 The authors justification for removing the Netherlands from the analysis is reasonable, but it is concerning to see the extent to which there was a gender imbalance in cases towards women. Under the model used in this paper, what is the probability that such a large deviation occurred due to random noise alone? Moreover, if this probability is small, is it possible that there is insufficient noise assumed in the model, and hence that some of the significance of the results in countries which saw a gender imbalance towards men has been over-stated?

We appreciate the feedback. As you have already noticed the opposite effect in the Netherlands is quite an interesting phenomenon. The observed effect is very likely due to the simultaneously occurring freedom day. We were told that with opening dancing locations (clubs), especially women made use of that opportunity - and hence got infected with higher probability. However, we have no scientific sources that confirm this effect around the freedom day. It is also well possible that further phenomena affected the male and female population in a different way.

Nevertheless, we can estimate the probability that such an event solely occurred due to random noise under our model. Our model estimates the noise on the gender imbalance $\sigma_{\Delta\tilde{\gamma}}$ to be about 0.02 (95% CI: [0.007, 0.06]). In order to obtain an imbalance such that the deviation can be explained, one needs a change of ΔR_{noise} of about 0.17. Using the upper estimate of the $\sigma_{\Delta\tilde{\gamma}}$, such a change requires a deviation of about 2.8σ , which corresponds to a (two-sided) probability of 0.5%.

Hence, we do not interpret this occurrence as an event due to random noise as parameterized in R_{noise} . However, in this case, your follow-up question on the possible occurrence of such events at other times during the championship is of high relevance.

Therefore, we perform the following test: Assuming the noise to be under-represented in the model, a counterfactual shift of the date of the matches would lead to the random occurrence of some significant results in the effect size. This is not the case for delays of 14, ± 30 , ± 35 , ± 40 days (see Supplementary Figure S11 and S12). Here, the counterfactual results always include an effect size of zero within a 95% CI in contrast to the factual result (The largest deviation at 35 days is at the 93.7th percentile). Hence, we can exclude that the significance of the result is due to under-represented noise. Please also note that the offset results do not all show the exact same result, but do vary within the credible interval of the result. This is expected since the model can randomly attribute variations of case numbers and gender imbalances on a timescale of the average time span between games or shorter to R_{soccer} . The indicative agreement between the range of variation between offset results and the CI in this low statistics sample of 6 experiments hints at the correctness of the statistical result.

Related to this discussion is answer 2 to Reviewer 1. There we show the variance of the changes in the gender imbalance in two time slices: once in the 35 days directly before the championship, and once in the 30 days of the championship plus one generation interval of 5 days for all analyzed countries. One observes that the variance is larger during the championship in most countries, supporting the hypothesis that the gender imbalance varied more than usually during the championship.

3 In equations (22) and (31), it appears that the sums are over all n - assumedly the sum should only be over those n such that the changepoint associated with n has already occurred?

Thank you for this question. It pointed us to an error in equation (25): Instead of

$$\Delta\gamma_n \sim \mathcal{N}(\Delta\gamma_{n-1}, \sigma_{\Delta\gamma}) \quad \forall n$$

it should have been

$$\Delta\gamma_n \sim \mathcal{N}(0, \sigma_{\Delta\gamma}) \quad \forall n.$$

We also added an explanatory sentence to these equations (lines 436ff):

“The idea behind this parameterization is that $\Delta\gamma_n$ models the change of R-value, which occurs at times d_n . These changes are then summed in equation (24). Change points that have not occurred yet at time t do not contribute in a significant way to the sum as the sigmoid function tends to zero for $t \ll d_n$.”

We hope this makes our choice in the equations a bit clearer.

4 Prior distributions are chosen throughout (e.g. equations (9), (16), (21)) without justification. To what extent are the results dependent on these choices? It is important that the sensitivity of the conclusions is explored sufficiently to inform readers (and indeed referees) of their strength. This should not be avoided citing environmental reasons.

Most priors are chosen to be rather uninformative, having little influence on our results. To show that, we multiplied the values of all those priors by a factor of 0.5 and 2 and show that the results do not change (Supplementary Fig. S18). In the methods text below, we added the respective equation when the robustness of the prior choices are investigated. The influence of equation (9) had already been investigated in Supplementary Fig. S16. In addition, for priors for which the parameterization makes it difficult to assess what the numbers represent, we added the 95% CI as information. Concretely this concerns equation (6) lines 408–412:

“ Here, we have the prior assumption that contacts between women, contacts between men, and contacts between women and men are equally probable. Hence, we chose the parameters for the Beta distribution such that c_{off} has a mean of 50% with a 2.5th and 97.5th percentile of [27%, 77%]. This prior is chosen such that it is rather uninformative. As shown in Supplementary Fig. S18, this and other priors of auxiliary parameters do not affect the parameter of interest if their width is varied within a factor of 2 up and down. ”

We also added the following text for equations (49) and (50) which parameterize the fraction of delayed cases during the week (lines 476 – 480).

“ We chose the prior of $r_{\text{base},d}^{\dagger}$ for Tuesday, Wednesday and Thursday such that only a small fraction of cases are delayed during the week. The chosen prior in equation (48) corresponds to a 2.5th and 97.5th percentile of r_d of [0%; 5%]. For the other days (Friday, Saturday, Sunday, Monday), the chosen prior leaves a lot of freedom, equation (49) corresponds to a 2.5th and 97.5th percentile of r_d of [0%; 72%]. ”

These three priors (eqs. (6), (49) and (50)) were excluded from the robustness analysis of Supplementary Fig. S18 because their parameterization makes it difficult to multiply it by a single number. However they encode reasonable assumptions. For instance, for equation (49) the assumption that cases do not have an additional delay during the week is the canonical choice. It would have been the same assumption if we would have chosen to model the delay on different weekdays in the same way.

5 If environmental concerns, as cited in Figures S9 and S10, are valid, then can the authors determine computational efficiencies that would make the investigation more feasible without unreasonable environmental costs?

We ran our robustness analyses (Supplementary Figures S13 to S18) for the missing countries, however with half the length of the MCMC chains. This reduces the quality of the posterior distribution estimation for these countries a little.

6 Would it be possible to run the same analysis, but to initialise the model significantly before the start of the championship? This would provide a good examination of the interaction between the base and noise terms.’

Thank you for the idea. We have run our analysis starting one month earlier, and added the figures to the manuscript (Supplementary Fig. S23 and S24). We can not see a significant difference in noise terms using the longer time period (see below).

Moreover the effect size is not altered in any significant way (see below).

7 Why are the p-values given for one-sided, rather than two-sided, tests? [For example, in Figure S6.] Indeed Figure S6C looks like the two-sided p-value should be considerably greater than 0.06. Also why is the central line so near the top of the shaded area? This would appear to be an error which makes me concerned about all of the graphs with such shading.

Thank you very much for noticing! We found a small error in the computation of the CI. Instead of the lower bound of 2.5% we computed the 0.25% bound. This is fixed now in all figures and is displayed correctly.

The tests are one-sided, because we don't exactly use p-values, but the Bayesian counterpart the "Probability of Direction" (see e.g. [Makowski2019]). It is simply the proportion of the posterior distribution that corroborates the hypothesis (or support the alternative hypothesis when used similarly to the p-value) and it is therefore one-sided.

8 Do the linear regression models plotted in Figure 3 (and elsewhere) allow for heteroscedasticity? If not, then this should be allowed for given the considerable variability in uncertainty associated with the plotted estimates. The methods section should make 100% clear what how linear regressions have been estimated and what is plotted.

The regression indeed has heteroscedastic errors since it includes the individual posterior uncertainties of the effect size individually for each data point. Beyond this, the Bayesian parameterization of the regression adds one parameter which models the consistency of the data with a linear dependence. It is chosen as a constant value for each entry, since it characterizes the applicability of the chosen functional dependence and is not a property of each measurement. We added the following explanation in the methods (lines 539–543):

“ Therefore our regression model includes the “measurement error” $\hat{\sigma}_c$ which models the heteroscedastic effect size of every country, and an additional model error τ which models the homoscedastic deviations of the country effect sizes from the linear model. In the plots, we plot the regression line \hat{Y}_c with its shaded 95% CI, and data points (\hat{X}_c, Y_c^\dagger) where the whiskers correspond to the one standard deviation, modeled here by ϵ_c and $\hat{\sigma}_c$. ”

Editorial comments:

E1 Be consistent with decimal / comma notation (e.g. 10.000 is used in the abstract to mean what is given as 10000 elsewhere)

Thank you for noticing the inconsistency. We now consistently use commas in the text.

E2 Line 29: There is potentially some link between pandemic state and team progression (e.g. Billy Gilmour was forced to isolate for Scotland).

This is a good point. We rewrote the corresponding paragraph in the introduction, stating that the relation has a random component (lines 27f):

“First, the Euro 2020 resembles a randomized study across countries: The time-points of the matches in a country do not depend on the state of the pandemic in that country and how far a team advances in the championship as a random component as well [20].”

However, we believe that this link will not be strong because most teams should have training (camps) before and during the tournament where additional tests and measures helped to partially decouple the pandemic situation from the home country. Social and physical distancing were strongly encouraged by relevant authorities, see e.g. the statement from UEFA Euro 2020 chief medical officer Dr Zoran Bahtijarevi and UEFA Euro 2020 medical advisor Dr Daniel Koch:

“If the teams respect our recommendations, they are actually travelling from their base camp, which is a bubble and which should be a protected environment. They will be travelling using their own group of vehicles in which all the drivers have been tested, and most of them are vaccinated too. They will fly on their own charter flight, and at the airport theyre using special boarding procedures, which is also actually limiting their contact with the population. When they arrive in a country, they have a special disembarkation procedure, they use their own vehicles and travel to a protected environment at the hotel. Id like to take this opportunity to congratulate the teams on qualifying for the last 16... and would call on them once again to respect the measures in place, because they are there for their benefit.”

A listing of the observed COVID-19 cases of players during the isolation period before the championship (see Reuters Article) displays sufficiently low statistics to assume no direct connection. Moreover, for three of the twelve teams, the team base was not in their home country.

Although all of this does not guarantee that there is no potential link between the pandemic state and team progression, it greatly reduces its likelihood.

We added a few sentences about this in the discussion (lines 184–189):

“ Our results might further be biased if the incidence and the teams’ progression in the Euro 2020 are correlated. It is conceivable that high incidence would negatively correlate with team progression through ill or quarantined team members. However, there were only few such cases during the Euro 2020 [Reuters Article], and the correlation might also be positive: At higher case numbers the team might be more careful. Hence, the correlation is unclear and probably negligible. ”

E3 Line 166: Should be FIFA World Cup instead of World Championship

Indeed, thank you. We have corrected it accordingly.

E4 Table S2, row 5: Typo - you have R_{base} instead of R_{base}

Thank you for noticing. We have corrected it.

E5 Line 389: $\alpha_{prior,m}$ is acting as a function of m rather than a matrix (also in Table S1)

We are sorry for the unclear notation and definition. We clarified the object as (line 423f):

“ $\alpha_{prior,m}$ is the m -th element of the vector that encodes the prior expectation of the effect of a match on the reproduction number. ”

E6 Why is $\beta_{prior,m}$ not listed in Table S1?

Thank you for noticing that we forgot to put in there. We have now listed it in the table.

E7 Inconsistency throughout between The Czech Republic and Czechia

Thank you for pointing us to the inconsistency. We have corrected all instances of “Czechia” to “Czech Republic”.

E8 In Figure S7, the terms Quarter-finals and Semifinal are fine, but there is only one Final it is not Finals. The terms are not quarter finale [Fig S19] or finale [Fig S15]. Further, the paper should not switch between final match and finals. The text (line 98) is confusing when it refers to final matches since there is only one final match.'

Thank you for pointing this out. We have corrected “finals” to “final”, “final matches” to “last matches of the championship” and “quarter finale” to “quarterfinals” in the text and also in Supplementary Figure S9.

E9 Line 442: Should be eq. (44) i.e. the brackets are needed.

Thank you for noticing. We corrected it (line 467).

E10 Table S10: Time in championship should really be time between first and last match, shouldnt it?

You are completely right. We rephrased the column name to “Time between first and last match of the country (days)” to clarify.

E11 In all cases the figure captions need to clearly define the shading as well as any lines that have been plotted. For example, it would appear that Fig S5 shows linear regression model estimates and 95% confidence intervals, but this is not stated.

Thank you for noticing this inconsistency. Throughout the manuscript, we always use 95%, 68% credible intervals or one standard deviation. We have added the missing definitions to the figure captions.

Reviewer 3

This study aims to quantify the impact of the UEFA Euro 2020 Football Championship on the spread of COVID-19 among 12 countries to influence public health policy. This is an interesting paper that exemplifies the importance of public health policies regarding large-scale sporting events. I found one major limitation in the estimation of the number of deaths associated with the analyzed events and a set of other relatively easily addressable points.

We thank you for your detailed comments and suggestions, which led us to improve our manuscript. Below we address them point by point.

1 Line 37, disease transmission rates → infection transmission rates. The disease cannot be transmitted; the infection (or the pathogen) is transmitted.

Thank you for pointing this out. According to the definition in [cancer.gov](https://www.cancer.gov), we understand an infection as a process, i.e. nothing that can be transmitted. We are glad about your second suggestion and corrected it to “pathogen transmission rates” (line 36f).

2 Line 47. Basic should be base (according to the nomenclature used in the rest of the manuscript).

Thank you for noticing. We have corrected it accordingly.

3 Line 64. Primary cases are defined as infections occurring at gatherings on match days. How are these primary cases identified? And how do you differentiate 1) between primary and subsequent cases and 2) cases that occur from different matches?

Thank you for noticing the lack of a definition of primary and subsequent cases. We added a subsection in the method section which reads (lines 497–501):

“ We compute the number of primary football related infected $I_{\text{primary},g}(t)$ as the number of infections happening at football related gathering. The percentage of primary cases f_g is then computed by dividing by the total number of infected $I_g(t)$.

$$I_{\text{primary},g}(t) = \frac{S(t)R_{\text{football}}(t)}{N} \sum_{g'} I_{g'}(t) C_{\text{football},g',g} \quad (2)$$

$$f_g = \sum_t \frac{I_{\text{primary},g}(t)}{I_g(t)} \quad t \in [\text{11th June, 31st July}] \quad (3)$$

To obtain the subsequent infected $I_{\text{subsequent},g}(t)$ we subtract infected obtained from a hypothetical scenario without football games $I_{\text{none},g}(t)$ from the total number of infected.

$$I_{\text{subsequent},g} = I_g(t) - I_{\text{primary},g}(t) - I_{\text{none},g}(t) \quad (4)$$

$$(5)$$

Specific, we consider a counterfactual scenario, where we sample from our model leaving all inferred parameters the same expect for the football related reproduction number $R_{\text{football},g}(t)$, which we set to zero. ”

4 In lines 66-67, you mention “We included all subsequent until July 31...”. Were subsequent cases for all participating countries analyzed until July 31 or was that only for the countries involved in the final match? If yes, how do you justify that countries participating only in early matches are still contributing to subsequent COVID-19 cases long after the matches? If all countries were not included until July 31, were subsequent cases two weeks after the countrys final match included in the analysis?

Thank you very much for these considerations. There is a lot of freedom of choice at that point. One could also argue that the time of the first match of each country should be the one that determines until when the subsequent cases are considered. For enhanced clarity, we decided to go with fixed dates. We are also comparing the absolute case and deaths numbers. For this comparison we need to use common fixed dates.

5 Line 78. First, it is SARS-CoV-2 infections and not COVID-19 infections. Second, these are reported SARS-CoV-2 infections, which are large underestimations of the true number of infections. Please rephrase and add a comment on this in the Discussion.

Thank you for pointing this out. We have corrected our wording where applicable, and discussed further on the effect of possible additional testing and reporting. In the discussion, lines 210–211 now read:

“However, we expect that some individuals would actively get tested right after a match, thereby increasing the case finding and reporting rates. ”

Moreover, in response to your comment **12** we emphasize the definition of a case in the introduction lines 29–41:

In the following, we use “case” to refer to a confirmed case of a SARS-CoV-2 infections in a human and “case numbers” to refer to the number of such cases. Not all infections are reported and represented in the cases and cases come with a delay after the actual infections.

A likely underestimation of the true pandemic state as seen only by confirmed cases does not alter the key findings of our analysis, since we attribute *observed* cases to championship-related fan activity and make no statement about additional unreported cases.

6 Line 79. First, that is a case fatality ratio, not a rate. Rates are expressed in $time^{-1}$, while you are using that as a ratio instead. Second, exactly as there is a gender imbalance in the population affected by Euro 2020, there very likely is an age imbalance as well. Specifically, we expect that population to be much younger than the general population of the country. As such, for a disease like COVID-19 where the fatality is much higher in the elderly, applying an age-independent case fatality ratio provides hardly credible results. I strongly encourage the authors to either to rely on age-dependent estimates of the case fatality ratio or to entirely drop the estimates of the number of deaths.

Thank you for pointing this out. We have replaced "case fatality rate" for "case fatality risk" everywhere in our manuscript, so that it is clear that it does not have units and it refers to the chances of an individual dying given infection.

Furthermore, the strong coupling in cases between age groups – exemplified by the lack of success in Sweden to protect the elderly in care homes (see bmj Article) – hints at a strong coupling of infections between age groups. Since the total football-related cases are dominated by subsequent cases, we can assume that the primary cases spread rapidly over age groups. We have added a word of caution to the manuscript in lines 83–87:

This is likely slightly overestimated because the age groups most at risk from COVID-19 related death are probably underrepresented in football-related social activities and thus more unlikely to be affected by primary championship-related infections. However, the overall number of primary and subsequent cases attributed to the championship is dominated by the subsequent cases, and the mixing and infections between age-groups then mitigates this bias.

We did also remove the estimate of the number of deaths from the abstract and the conclusion paragraph of the discussion to not overemphasize this result.

7 Connected to the previous point, it is possible that the case reporting rate has temporarily increased right after each match. This should be discussed as a study limitation.

Thank you for pointing this out. We have now included both this phenomenon when discussing testing before as well as after a match (and associated gatherings) and its implications for our results. Now lines 210–214 read:

“ However, we expect that some individuals would actively get tested right after a match, thereby increasing the case finding and reporting rates. This can slightly affect our estimates for the delay distribution D and would require additional information to be corrected. Altogether, analyzing large-scale events with precise timing and substantial impact on the spread presents a promising, resource-efficient complement to classical quantification of delays. ”

8 Line 122. A generation interval of 4 days appears to be very short. That could be a reasonable estimate for a Chinese setting with very isolation policies in dedicated facilities, but rather short for a European context with very loose household isolation policies. 6 days would be a more sensible choice (see for instance Manica et al, Estimation of the incubation period and generation time of SARS-CoV-2 Alpha and Delta variants from contact tracing data, Medrxiv).

Thank you for pointing us to this. We investigated the impact of a longer generation interval on our results and found out that the differences are negligible (Supplementary Fig. S17). It mainly changes the inferred base reproduction number, but the total impact of the championship remains mostly the same. Therefore we kept our current model as the base model.

9 Lines 145-147 and 148-150. These sentences are speculative. It might well be the case that such events should be entirely banned during certain epidemic phases and/or mass gatherings avoided altogether. Moreover, the authorities should not do anything based on a manuscript. Each authority should make the decision based on its specific targets and priorities (which may not be aligned with those considered in this manuscript).

Thank you, this is an important point. We have rephrased these sentences to be rather explanatory than demanding (lines 158–163):

“ To prevent the impacts of these events, measures, such as promoting vaccination, enacting mask mandates, and limiting gathering sizes, can be helpful. Besides, the effectiveness of such interventions has already been quantified in different settings (e.g., [Brauner2020, Sharma2021]) so that policymakers can weigh them according to specific targets and priorities. Furthermore, focused measures that aim to mitigate disease spread *in situ*, such as testing campaigns and requiring COVID passports to attend sport-related gatherings and viewing parties, present themselves as helpful options. ”

10 Line 150-151. I agree with this point, but it is phrased rather badly. The incubation period has a wide distribution, and its mean is not representative of the whole phenomenon. Moreover, not only the mean of the incubation period but also the mean of the generation time is in line with the interval between matches.

Thank you for pointing this out. We have rephrased the whole passage to be clearer. Now lines 164 – 168 read:

Moreover, the championship distribution of matches every 4 to 5 days coincides with the mean incubation period and generation interval of COVID-19. This means that individuals who get infected watching a match can turn infectious by the subsequent while potentially pre-symptomatic. Such resonance effects between gathering intervals and incubation time can increase the spread considerably [Zierenberg2021].

11 In Figures 2 and S4, base cases are named independent cases in the figure, but the captions and main text all refer to them as base cases. I suggest keeping these labels consistent throughout the paper and figures.

Thank you for noticing this. We have removed every instance of “base case” to restore consistency.

12 In general, there is quite a bit of confusion between cases and infections that the authors appear to be used interchangeably, while they are two clearly defined and different epidemiological concepts. Please carefully revise the wording throughout the manuscript.

Thank you for noticing this inconsistency. As mentioned in our reply to your comment 5, we added a remark in the introduction (lines 39–41), which reads:

In the following, we use “case” to refer to a confirmed case of a SARS-CoV-2 infections in a human and “case numbers” to refer to the number of such cases. Not all infections are reported and represented in the cases and cases come with a delay after the actual infections.

Additionally, we replaced instances of infections with cases where we found it to be more accurate.

REVIEWERS' COMMENTS

Reviewer #1 (Remarks to the Author):

The authors have satisfactorily answered to my comments.

Reviewer #2 (Remarks to the Author):

It is useful that Figure S8 has been added to demonstrate the impact of excluding England and Scotland. However, I don't feel that this text:

"The strong significance of this correlation relies mainly on England and Scotland. However, the observed trend in an analysis without these two countries, while not significant at the 95% confidence level, is consistent with the findings including all countries. This is shown in supplementary Fig. S8."

puts enough information into the main text. The R^2 is 0.09 when England and Scotland are excluded, with a 95% CI of (0.00, 0.49). It would be useful if the slope estimates and 95% CIs were given in the text as well - I only know them because they were included in the response: "However, the regression parameters are consistent between all data points (1.62 [1.0, 2.26]) and the result without the UK (0.76 [-1.46, 3.04])."

This statement makes clear that without England and Scotland, there is almost no information.

Finally, it is not helpful that the x-axis numbers for Fig S8 are "500", "1k", "2k" and again "2k". There is plenty of space to make this figure larger so 500, 1000, 1500 and 2000 can be used rather than having "2k" representing both 1500 and 2000.

Reviewer #3 (Remarks to the Author):

The authors have adequately addressed my comments. Please find below a short list of very minor comments.

Line 44: "infections" -> "infection"

Line 94: "[...] mixing of infections between age-groups then [...]" -> "[...] mixing between individuals of different age groups then [...]"

Line 408: Ref. 46 does not provide estimates of the generation interval.

Lines 408 and 409: "[...] but shorter than the estimated serial interval of the original strain.". First, a reference is missing here. Second, why do the authors refer to the serial interval here since estimates of the generation interval for the ancestral lineages of SARS-CoV-2 are available in the literature? See for instance, <https://www.nature.com/articles/s41467-021-21710-6> and <https://www.science.org/doi/full/10.1126/science.abb6936> .

December 2, 2022

Revision of our manuscript to *Nature Communications*

Dear reviewers,

thank you very much for all the helpful comments during the review period. We addressed the last comments as detailed below.

Viola Priesemann and Philip Bechtle
(on behalf of all authors)

Reviewer 1

The authors have satisfactorily answered to my comments.

Thank you again for your helpful comments.

Reviewer 2

It is useful that Figure S8 has been added to demonstrate the impact of excluding England and Scotland. However, I don't feel that this text: "The strong significance of this correlation relies mainly on England and Scotland. However, the observed trend in an analysis without these two countries, while not significant at the 95% confidence level, is consistent with the findings including all countries. This is shown in supplementary Fig. S8." puts enough information into the main text. The R^2 is 0.09 when England and Scotland are excluded, with a 95% CI of (0.00, 0.49). It would be useful if the slope estimates and 95% CIs were given in the text as well - I only know them because they were included in the response: "However, the regression parameters are consistent between all data points (1.62 [1.0, 2.26]) and the result without the UK (0.76 [-1.46, 3.04])." This statement makes clear that without England and Scotland, there is almost no information.

Indeed, it is helpful to be more informative here. We added to the main text the slope estimates:

" Indeed, we find a clear correlation between the observed and the expected incidence Fig. ??a, $R^2 = 0.77$ (95% CI [0.39,0.9]), $p < 0.001$, with a slope of 1.62 (95% CI [1.0, 2.26]). The strong significance of this correlation relies mainly on England and Scotland. However, the observed slope in an analysis without these two countries (0.76, 95% CI: [-1.46, 3.04]), while not significant at the 95% confidence level, is consistent with the findings including all countries. This is shown in supplementary Fig. S7. "

And also added these slope estimate to the caption of supplementary Fig. S7 and emphasized that the correlation is not significant:

" The potential for spread, i.e., the number of COVID-19 cases that would be expected during the time T a country is playing in the Euro 2020 ($N_0 \cdot R_{pre}^{T/4}$) is still correlated with the number of Euro 2020-related cases after removing the two most significant entries from the analysis but not significantly. The observed slope without the most significant countries (median: 0.76, 95% CI: [-1.46, 3.04]) is consistent within its uncertainties with the slope including all countries (median: 1.62, 95% CI: [1.0, 2.26]). "

Finally, it is not helpful that the x-axis numbers for Fig S8 are "500", "1k", "2k" and again "2k". There is plenty of space to make this figure larger so 500, 1000, 1500 and 2000 can be used rather than having "2k" representing both 1500 and 2000.

Thank you for spotting this. We corrected the x-axis numbers as suggested.

Reviewer 3

Line 44: “infections” -> “infection”

Line 94: “[...] mixing of infections between age-groups then [...]” -> “[...] mixing between individuals of different age groups then [...]”

We corrected it as suggested.

Line 408: Ref. 46 does not provide estimates of the generation interval.

Lines 408 and 409: “[...] but shorter than the estimated serial interval of the original strain.”. First, a reference is missing here. Second, why do the authors refer to the serial interval here since estimates of the generation interval for the ancestral lineages of SARS-CoV-2 are available in the literature? See for instance, <https://www.nature.com/articles/s41467-021-21710-6> and <https://www.science.org/doi/full/10.1126/science.abb6936> .

Indeed, the literature references are not suitable, and referring to the serial interval instead of the generation interval isn't appropriate. We replaced the previous reference 46 which only estimated the serial interval by [47] W. S. Hart, et al. (2022) ([https://www.thelancet.com/journals/laninf/article/PIIS1473-3099\(22\)00001-9](https://www.thelancet.com/journals/laninf/article/PIIS1473-3099(22)00001-9)) and added the two proposed references for the generation interval of the original strain ([48, 49]). We also added here a reference to the robustness check figure of the generation interval:

“ This generation interval (between infections) is modeled by a Gamma distribution $G(\tau)$ with a mean μ of four days and standard deviation σ of one and a half days. This is a little longer than the estimates of the generation interval of the Delta variant [45, 46], but shorter than the estimated generation interval of the original strain [47, 48]. The impact of the choice of generation interval has negligible impact on our results (supplementary Fig. S7). ”